# Convergence of monosynaptic and polysynaptic sensory paths onto common motor outputs in a *Drosophila* feeding connectome

Anton Miroschnikow[1], Philipp Schlegel[1,2], Andreas Schoofs[1], Sebastian Hueckesfeld[1], Feng Li[3], Casey M Schneider-Mizell[3], Richard D Fetter[3,4], James W Truman[3], Albert Cardona[3,5], Michael J Pankratz[1]*

[1]Department of Molecular Brain Physiology and Behavior, LIMES Institute, University of Bonn, Bonn, Germany; [2]Department of Zoology, University of Cambridge, Cambridge, United Kingdom; [3]Janelia Research Campus, Howard Hughes Medical Institute, Ashburn, United States; [4]Department of Biochemistry and Biophysics, University of California, San Francisco, San Francisco, United States; [5]Department of Physiology, Development and Neuroscience, University of Cambridge, Cambridge, United Kingdom

**Abstract** We reconstructed, from a whole CNS EM volume, the synaptic map of input and output neurons that underlie food intake behavior of *Drosophila* larvae. Input neurons originate from enteric, pharyngeal and external sensory organs and converge onto seven distinct sensory synaptic compartments within the CNS. Output neurons consist of feeding motor, serotonergic modulatory and neuroendocrine neurons. Monosynaptic connections from a set of sensory synaptic compartments cover the motor, modulatory and neuroendocrine targets in overlapping domains. Polysynaptic routes are superimposed on top of monosynaptic connections, resulting in divergent sensory paths that converge on common outputs. A completely different set of sensory compartments is connected to the mushroom body calyx. The mushroom body output neurons are connected to interneurons that directly target the feeding output neurons. Our results illustrate a circuit architecture in which monosynaptic and multisynaptic connections from sensory inputs traverse onto output neurons via a series of converging paths.
DOI: https://doi.org/10.7554/eLife.40247.001

*For correspondence:
pankratz@uni-bonn.de

Competing interests: The authors declare that no competing interests exist.

## Introduction

Motor outputs of a nervous system can be broadly defined into those carried out by the muscles to produce movements and by the glands for secretion (*Shepherd, 1987*). Both of these behavioral and physiological events are regulated by a network of output neurons, interneurons and sensory neurons, and a major open question is how one neural path is selected from multiple possible paths to produce a desired output (*Grillner et al., 2005*). Nervous system complexity and tool availability have strongly dictated the type of experimental system and analysis that can be used to address this issue, such as a focus on a particular organism, behavior or type of neuron. In this context, the detailed illustrations of different parts of nervous systems at neuronal level as pioneered by Cajal, to the first complete description of a nervous system wiring diagram at synaptic level for *C. elegans*, demonstrate the power of systematic neuroanatomical analysis in providing a foundation and guide for studying nervous system function (*Ramón y Cajal, 1894*; *White et al., 1986*). However, the technical challenges posed by such analysis have limited the type of organisms for which synaptic

resolution mapping can be performed at the scale of an entire nervous system (*Swanson and Lichtman, 2016*; *Schlegel et al., 2017*; *Kornfeld and Denk, 2018*).

Analysis of the neural circuits that mediate food intake in the *Drosophila* larvae offers numerous advantages in meeting the challenge of neuroanatomical mapping at a whole brain level, and combining it with the ability to perform behavioral and physiological experiments. The muscle system that generates the different movements necessary for transporting food from the pharynx to the esophagus, as well as the endocrine system responsible for secreting various hormones for metabolism and growth, have both been well described (*Kühn, 1971*; *Siegmund and Korge, 2001*; *Buch and Pankratz, 2009*; *Schoofs et al., 2010*). These are also complemented by the analysis of feeding behavior in adult flies (*Gelperin, 1971*; *Dethier, 1976*; *McKellar, 2016*). Although there is broad knowledge at the morphological level on the organs underlying larval feeding behavior and physiology, as well as on the nerves innervating them in the periphery (*Schoofs et al., 2010*; *Schoofs et al., 2014b*), the central connectivity of the afferent and efferent neurons within these nerves are largely unknown. At the same time, advances in the EM reconstruction of an entire CNS of a first instar larva (*Ohyama et al., 2015*; *Schlegel et al., 2016*; *Schneider-Mizell et al., 2016*; *Berck et al., 2016*; *Eichler et al., 2017*; *Gerhard et al., 2017*) (summarized in *Kornfeld and Denk, 2018*) offers an opportunity to elucidate an animals' feeding system on a brain-wide scale and at synaptic resolution. As part of this community effort, we recently performed an integrated analysis of fast synaptic and neuropeptide receptor connections for an identified cluster of 20 interneurons that express the neuropeptide hugin, a homolog of the mammalian neuropeptide neuromedin U, and which regulates food intake behavior (*Melcher et al., 2006*; *Schoofs et al., 2014a*; *Schlegel et al., 2016*). This analysis showed that the class of hugin neurons modulating food intake receives direct synaptic inputs from a specific group of sensory neurons, and in turn, makes monosynaptic contacts to output neuroendocrine cells. The study not only provided a starting point for a combined approach to studying synaptic and neuropeptidergic circuits (*Diao et al., 2017*; *Williams et al., 2017*), but a basis for a comprehensive mapping of the sensory and output neurons that innervate the major feeding and endocrine organs.

Feeding is one of the most universal and important activities that animals engage in. Despite large differences in the morphology of the external feeding organs, the internal gut structures are quite similar across different animals (*Campbell, 1990*); indeed, even within closely related species, there can be large differences in the external organs that detect and gather food, whereas the internal organs that transport food through the alimentary canal are much more similar. Recent studies have also pointed out the functional similarities between the subesophageal zone in insects and the brainstem in vertebrates for regulating feeding behavior (*Schoofs et al., 2014a*; *Yapici et al., 2016*; *McKellar, 2016*). In mammals, the different cranial nerves from the medulla innervate distinct muscles and glands of the foregut (*Figure 1A*). For example, the VIIth cranial nerve (facial nerve) carries taste sensory information from anterior 2/3 of the tongue, and innervates the salivary glands, and lip and facial muscles. The IXth cranial nerve (glossopharyngeal nerve) receives taste inputs from the posterior 1/3 of the tongue, and innervates the salivary glands and pharynx muscles. The Xth cranial nerve (vagus nerve) receives majority of the sensory inputs from the enteric nervous system of the gut, and innervates pharynx and esophagus muscles. The XIth cranial nerve (spinal accessory nerve) and the XIIth cranial nerve (hypoglossal nerve) are thought to carry strictly motor information which innervate the pharynx and neck muscles, and the tongue muscles (*Cordes, 2001*; *Simon et al., 2006*). The distinct cranial nerves project onto topographically distinct areas in the medulla of the brainstem (*Figure 1A*). We also note that olfactory information is carried by cranial nerve I, a strictly sensory nerve that projects to the olfactory bulb (OB), an area topographically distinct from the brainstem. In addition, there are direct neuronal connections between the brainstem and the hypothalamus, the key neuroendocrine center of vertebrates (*D'Agostino et al., 2016*; *Liu et al., 2017*).

Analogously, distinct pharyngeal nerves of the *Drosophila* larva are connected to the subesophageal zone (SEZ), and also carry sensory and motor information that regulate different parts of the body (*Figure 1B*). The AN (antennal nerve) carries sensory information from the olfactory, pharyngeal and internal organs, and innervates the pharyngeal muscles for pumping in food. The serotonergic neurons that innervate the major endocrine center and the enteric nervous system also project through the AN (*Huser et al., 2012*; *Schoofs et al., 2014b*). Note also that the olfactory sensory organs project to the antennal lobe (AL), which abuts the SEZ yet is topographically separate. The

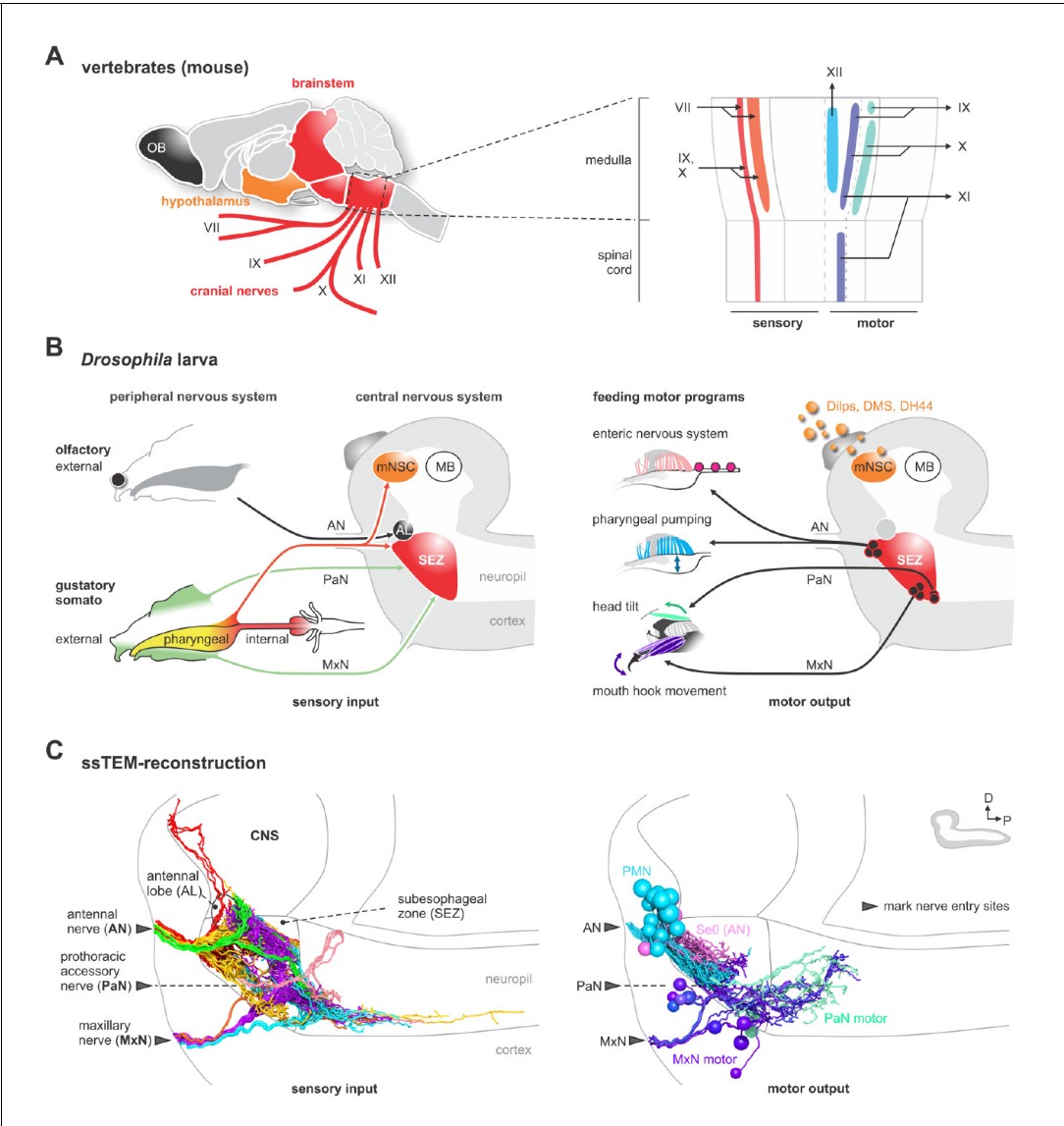

**Figure 1.** EM reconstruction of the pharyngeal nerves of *Drosophila* larva. (**A**) Left: schematic diagram shows a lateral view of an adult mouse brain and the broad organization of different cranial nerves targeting the medulla of the brainstem. Right: topographical chart of the medulla and part of the spinal cord. Primary sensory and primary motor nuclei are shown on the left and on the right, respectively. (**B**) Schematic overview of external, pharyngeal and internal sensory systems targeting the subesophageal zone (SEZ), median neurosecretory cells (mNSCs) and the antennal lobe (AL) in *Drosophila* (left panel). Schematic overview of central output neurons targeting feeding related muscles and the enteric nervous system (right panel). Median neurosecretory cells (mNSCs) target neuroendocrine organ and the periphery, by releasing neuropeptides such as Dilps, DMS and DH44. The mushroom body (MB), a learning and memory center, serves as a landmark. (**C**) EM reconstruction of pharyngeal sensory input (left panel). Sensory neurons enter the brain via the antennal nerve (AN), maxillary nerve (MxN) and prothoracic accessory nerve (PaN), and cover large parts of the SEZ (left panel). Arrowheads mark respective nerve entry site. Two of the AN sensory projections (per side) extend into the protocerebrum. EM reconstruction of pharyngeal motor output (right panel). Pharyngeal motor neurons (PMNs) and serotonergic output neurons (Se0) leave the CNS via the antennal nerve (AN) and innervate the cibarial dilator musculature (for pharyngeal pumping) and part of the esophagus and the enteric nervous system. MxN motor neurons leave the CNS via the maxillary nerve (MxN) and innervate mouth hook elevator and depressor, labial retractor and salivary gland ductus opener. PaN motor neurons leave the CNS via the prothoracic accessory nerve (PaN) and innervate the dorsal protractor (for head tilt movements). All neurons are colored based on their morphological class. See *Figure 1—figure supplements 1–4* and *Figure 2—figure supplement 6* for detailed anatomy and description of morphological clustering.

DOI: https://doi.org/10.7554/eLife.40247.002

The following source data and figure supplements are available for figure 1:

**Source data 1.** Summary of nerve nomenclature of *Drosophila melanogaster* larva.

DOI: https://doi.org/10.7554/eLife.40247.007

*Figure 1 continued on next page*

*Figure 1 continued*

**Figure supplement 1.** Anatomy of the pharyngeal nerves.
DOI: https://doi.org/10.7554/eLife.40247.003
**Figure supplement 2.** Sensory neurons of the antennal nerve.
DOI: https://doi.org/10.7554/eLife.40247.004
**Figure supplement 3.** Sensory neurons of the maxillary nerve.
DOI: https://doi.org/10.7554/eLife.40247.005
**Figure supplement 4.** Sensory neurons of the prothoracic accessory nerve.
DOI: https://doi.org/10.7554/eLife.40247.006

MxN (maxillary nerve) carries external and pharyngeal sensory information, and innervates the mouth hooks, whose movements are involved in both feeding and locomotion. The PaN (prothoracic accessory nerve) carries external sensory information from the upper head region, and innervates the muscles involved in head tilting (see *Figure 1—figure supplement 1* and *Figure 1—source data 1* for anatomical details and terminology). Furthermore, the SEZ has direct connections to median neurosecretory cells (mNSCs) and the ring gland. In sum, although a large body of knowledge exists on the gross anatomy of the nerves that target the feeding organs in vertebrates and invertebrates, the synaptic pathways within the brain that interconnect the sensory inputs and output neurons of the individual nerves remain to be elucidated.

In this paper, we have reconstructed all sensory, serotonergic modulatory (Se0) and motor neurons of the three pharyngeal nerves that underlie the feeding motor program of *Drosophila* larvae. The activity of these nerves has previously been shown to be sufficient for generating the feeding motor pattern in isolated nervous system preparations, and that the central pattern generators (CPGs) for food intake lie in the SEZ (*Schoofs et al., 2010*; *Hückesfeld et al., 2015*). We then identified all monosynaptic connections between the sensory inputs and the motor, Se0 and previously described median neurosecretory ouput neurons (*Schlegel et al., 2016*), thus providing a full monosynaptic reflex circuit for food intake. We also mapped polysynaptic pathways that are integrated onto the monosynaptic reflex circuits. In addition, we mapped the multisynaptic non-olfactory neuron connections from the sensory neurons to the mushroom body memory circuit (*Eichler et al., 2017*), and show that these are different from those involved in monosynaptic reflex circuits. Finally, we traced a set of mushroom body output neurons onto the neurosecretory and other feeding output neurons. Reflex circuits can be seen to represent the simplest synaptic architecture in the nervous system, as formulated by Charles Sherrington (*Sherrington, 1906*). Anatomical reconstructions of monosynaptic and polysynaptic reflex circuits can also be seen in the works of Cajal (*Ramón y Cajal, 1894*; *Swanson, 2000*). We propose a model of how different mono- and polysynaptic pathways can be traversed from a set of sensory neurons to specific output neurons, which has relevance for understanding the mechanisms of action selection.

## Results

### EM reconstruction of the pharyngeal nerves

We reconstructed all axons within the three pharyngeal nerves into the CNS using a ssTEM volume of an entire larval CNS (*Ohyama et al., 2015*) (*Figure 1C*). The sensory projections were those that ended blindly, whereas the motor and modulatory neurons were those with somata in the CNS. For sensory inputs, a regionalization of the target areas can already be seen, reflecting the fact that the nerves are fusions of different axon bundles that arise during embryonic development (*Hartenstein et al., 2018*; *Kendroud et al., 2018*). For example, only the AN has sensory projections that extend into the protocerebrum, whereas a major part of the MxN sensory projections extend into the ventral nerve cord. For motor neurons, the somata from the different nerves also occupy distinct regions within the SEZ (*Figure 1C*; *Figure 1—figure supplements 1–4*; *Figure 1—source data 1* for details on individual nerves, bundles and terminology).

## Topographical patterns of sensory and output synaptic compartments in the CNS

We next annotated all pre- and postsynaptic sites of all sensory projections and clustered them based on synapse similarity (*Figure 2A*). This revealed seven topographically distinct compartments in the CNS (*Figure 2B*; *Figure 2—figure supplements 1–2*). These compartments differ in the number of sensory neurons as well as in the identity of the pharyngeal nerve that gives rise to them (*Figure 2—figure supplements 3–5*). For example, the ACa ('anterior part of the Anterior Central sensory compartment') comprises 30 neurons that are exclusively derived from the AN; by contrast, the VM ('Ventromedial sensory compartment') comprises 102 neurons that derive from all three pharyngeal nerves. For neuroendocrine output neurons, we previously reconstructed the three neurosecretory cell clusters of the pars intercerebralis that innervate the major endocrine organ of *Drosophila* larvae (the ring gland). These express the neuropeptides Dilps (*Drosophila* insulin-like peptides), DH44 (diuretic hormone, a corticotropin releasing hormone homolog) and DMS (Dromyosuppressin), and receive monosynaptic inputs from the sensory system (*Figure 2C,D*; *Figure 3*) (*Schlegel et al., 2016*). We now identify here all pre- and post-synaptic sites of all the motor and modulatory neurons of the different pharyngeal nerves (*Figure 2C*). This includes a special class of four serotonergic neurons (the Se0 cluster) that project to the entire enteric nervous system (*Huser et al., 2012*; *Schoofs et al., 2014b*; *Shimada-Niwa and Niwa, 2014*). These four serotonergic neurons can be further divided into one that projects anteriorly to the pharynx (Se0ph), and three that project posteriorly towards the enteric nervous system (Se0ens) (*Figure 2—figure supplement 6*).

A schematic summary of the pre- and post-synaptic compartments of the input and output neurons, along with their projection regions, is shown in *Figure 2D*. Taken together, these data define all sensory input convergence zones and output compartments of the three pharyngeal nerves underlying feeding motor program at synaptic resolution.

## Axo-dendritic connections from sensory to neuroendocrine, modulatory and motor neurons

Having annotated all central synapses of in- and output neurons, we surprisingly found the most basic element of circuit architecture: direct monosynaptic connections between input and output neurons (*Figure 3A*). The vast majority of the monosynaptic connections are made from anterior 3 of the 7 sensory compartments (ACa, AVa, AVp; *Figure 3B,C*; *Figure 3—figure supplement 2*): around 90% of the neurons in these three compartments make monosynaptic contacts. Importantly, in- and output compartments do not perfectly overlap. As a consequence, we find single input compartments to make monosynaptic connections to neurons in overlapping output compartments, as one progresses from neuroendocrine (mNSCs), serotonergic neuromodulatory (Se0), and pharyngeal motor neurons (PMNs) (*Figure 3D*; *Figure 3—figure supplements 2–5*): ACa inputs onto the neuroendocrine and Se0 neurons; AVa onto neuroendocrine, Se0 and PMNs; AVp onto Se0 and PMNs; VM onto MxN and PaN motor neurons. Thus, the monosynaptic connections essentially cover all output neurons in contiguous, overlapping domains. When viewed from the sensory neuron side, a small percentage (less than 5% of synapses) makes monosynaptic contacts (*Figure 3E* left panel); however, when viewed from the output neuron side, the percentage of monosynaptic inputs they receive are substantial (*Figure 3E* right panel). For example, around 40% of all input synapses onto the serotonergic Se0 neurons, and between 10–25% of all mNSC input synapses, are from sensory neurons (*Figure 3F*; *Figure 3—figure supplements 2–5*). In sum, these results indicate that the monosynaptic connections between sensory and output neurons form a special class with which a core or an 'elemental' feeding circuit can be constructed (*Figure 3C*; *Figure 3—figure supplement 1*).

## Axo-axonic connections between sensory neurons

Unexpectedly we found a high number of synaptic connections between the sensory projections within the CNS. This is in contrast to the well characterized olfactory receptor neurons (ORNs) that project onto the antennal lobe (AL) (*Figure 4A*). For example, at a threshold of two synapses the AL has none, whereas 50% of the ACa neurons have above threshold inter-sensory connections. The majority of the inter-sensory connections were between neurons of the same synaptic target compartment (*Figure 4B*), which underscores the clear-cut boundaries between the sensory

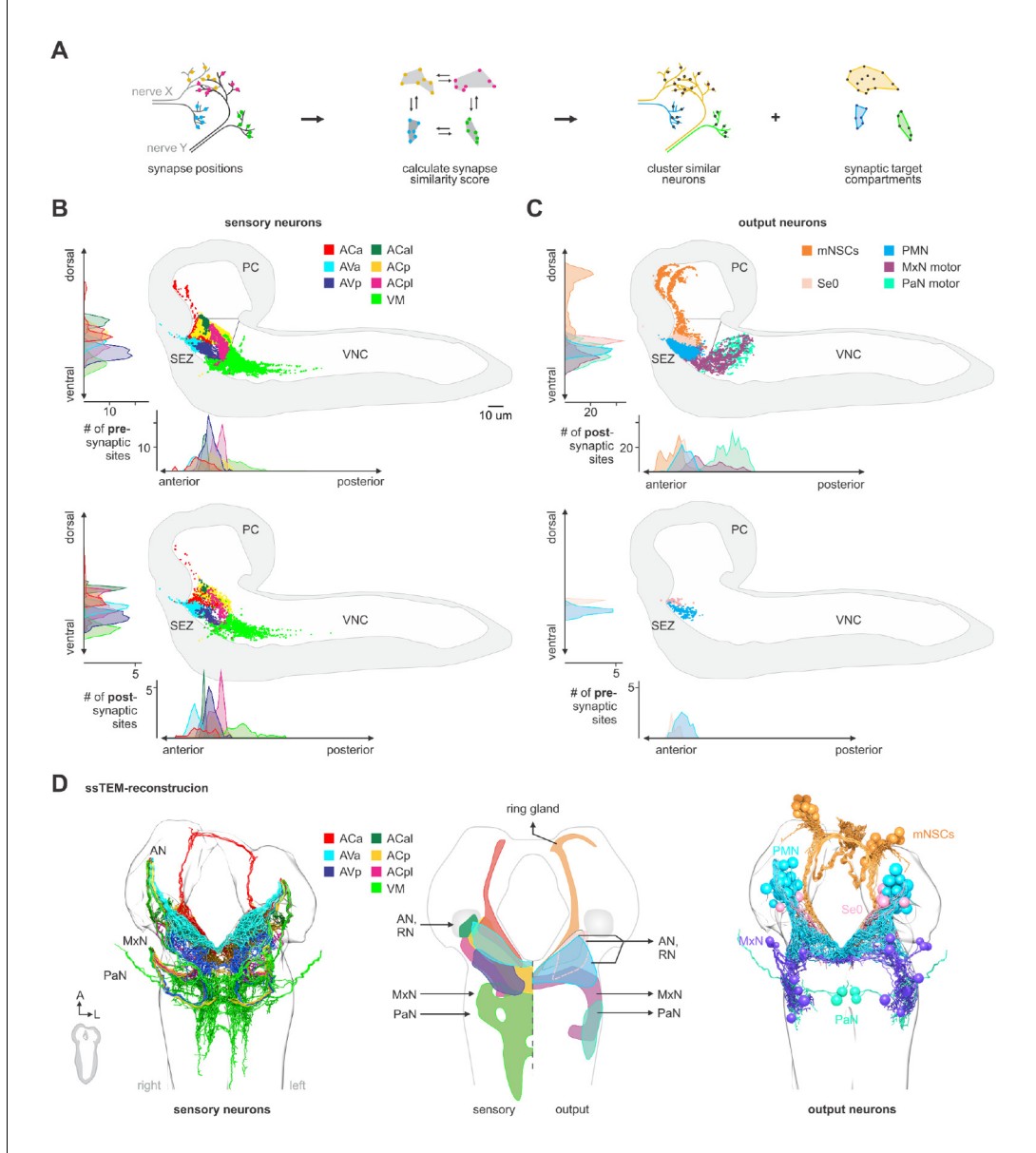

**Figure 2.** Spatially segregated central axonal projections of sensory neurons. (**A**) Calculation of pairwise synapse similarity score for all non-olfactory sensory neurons. (**B**) Spatial distribution of synaptic sites for all sensory neuron cluster. Hierarchical clustering based on synapse similarity score revealed seven distinct (non-overlapping) areas of sensory convergence within the SEZ: the anterior part of the Anterior Central sensory compartment (ACa), anterior part of the Anterior Ventral sensory compartment (AVa), posterior part of the Anterior Ventral sensory compartment (AVp), posterior part of the Anterior Central sensory compartment (ACp), anterior-lateral part of the Anterior Central sensory compartment (ACal), posterior-lateral part of the Anterior Central sensory compartment (ACpl) and Ventromedial sensory compartment (VM). Convergence zones are targeted by varying numbers of sensory neurons but are consistent across hemispheres. Each dot represents a single synaptic site. Graphs show distribution along dorsal-ventral and anterior-posterior axis of the CNS. (**C**) Spatial distribution of synaptic sites for all neuroendocrine, serotonergic and motor neuron classes. Each dot represents a single synaptic site. Graphs show distribution along dorsal-ventral and anterior-posterior axis of the CNS. (**D**) EM reconstruction of clustered sensory neurons (left). EM reconstruction of all output neuron classes (right). Summarizing representation of glomerular-like sensory compartments and motor compartments within the SEZ (middle panel). See *Figure 2—figure supplements 1–5* for detailed description of clustering and sensory region composition.

DOI: https://doi.org/10.7554/eLife.40247.008

The following figure supplements are available for figure 2:

**Figure supplement 1.** Similarity of sensory neuron synapse placement.

DOI: https://doi.org/10.7554/eLife.40247.009

*Figure 2 continued*

**Figure supplement 2.** Sensory compartments.
DOI: https://doi.org/10.7554/eLife.40247.010
**Figure supplement 3.** EM reconstruction of compartment forming sensory neurons and their synaptic sites.
DOI: https://doi.org/10.7554/eLife.40247.011
**Figure supplement 4.** EM reconstruction of compartment forming sensory neurons and their synaptic sites.
DOI: https://doi.org/10.7554/eLife.40247.012
**Figure supplement 5.** EM reconstruction of compartment forming sensory neurons and their synaptic sites.
DOI: https://doi.org/10.7554/eLife.40247.013
**Figure supplement 6.** Identification of serotonergic output neurons (Se0) in the EM volume.
DOI: https://doi.org/10.7554/eLife.40247.014

compartments; these connections are made both in an hierarchical manner as well as reciprocally, suggesting that sensory information processing is occurring already at an inter-sensory level in the brain (*Figure 4—figure supplement 1*). Viewed from output synapses of the sensory neurons, the percentage of sensory synapses connecting to other sensory neurons are small relative to total sensory outputs (less than 2% of 73,000 synapses); however viewed from input side of the sensory neurons, a high percentage (e.g., 45% for ACa) of their total synaptic inputs originate from other sensory neurons (*Figure 4C,D*). We also note that sensory neurons from ACp and VM have inter-sensory connections even between neurons of different nerves (*Figure 4—figure supplements 1–2*), indicating integration of sensory information from different body regions at the sensory neuron level.

## Mapping peripheral origins of monosynaptic circuits

We next investigated the peripheral origins of the sensory neurons that comprise the different synaptic compartments. This was accomplished by using various sensory receptor Gal4 lines to follow the projections from the sensory organs into the CNS. The mapping was aided by the fact that the pharyngeal projections enter the SEZ in distinct axon bundles that can be observed in both light and EM microscopic sections (*Figure 5A,B*). The AN and the MxN each have three bundles (these nerves are formed by fusion of several axon bundles during development) (*Hartenstein et al., 2018*), whereas the PaN has just one. The well characterized projections from the external olfactory organ (DOG) to the antennal lobe (AL), for example, use one of the bundles in the AN (Bundle 3 of the AN). *Figure 5B* illustrates the basic strategy, using two of the gustatory receptors (GRs) to follow the projections from the enteric nervous system into the CNS. This analysis, denoting the receptor line used and their expression in the sensory organs and the axon bundles of each pharyngeal nerve, is summarized in *Figure 5C* (*Figure 5—figure supplements 1–9* for detailed stainings). These results were then used to determine the peripheral origin (enteric/internal, pharyngeal, external) of the sensory neurons that comprise the seven synaptic compartments defined earlier (*Figure 5C,D*). This revealed a wide spectrum in compartment composition. For example, the ACa is derived 100% from the enteric nervous system, while the AVa is 93% enteric; these are the only two sensory compartments with enteric origin. As a comparison, the antennal lobe (AL) is derived 100% from a single external sensory organ, the dorsal organ. Interestingly, the topographical location of the sensory compartments within the CNS broadly mirrors in a concentric manner the peripheral origins from which they derive: the inner-most enteric organs project to the anterior most region, the pharyngeal sensory organs project to the middle region, while the most external organs project to the outer-most region (*Figure 5E*). Recent light microscopy study on the projections of somatosensory neurons onto the adult brain also showed topographically separate target areas in the brain (*Tsubouchi et al., 2017*).

In addition, as we progress from the inner to the outer layers, there is a graded contribution of connections having monosynaptic sensory-to-output contacts (highest being between the inner layers). In other words, the greatest number of monosynaptic connections occur between the enteric system and the neuroendocrine system, followed by the pharyngeal sensory neurons to the pharyngeal motor neurons, and the least from the external organs. We point out, for example, that the olfactory projections from the external dorsal organ have no monosynaptic connections whatsoever to any output neurons. In this context, the Se0 serotonergic neurons appear to play a special role, as

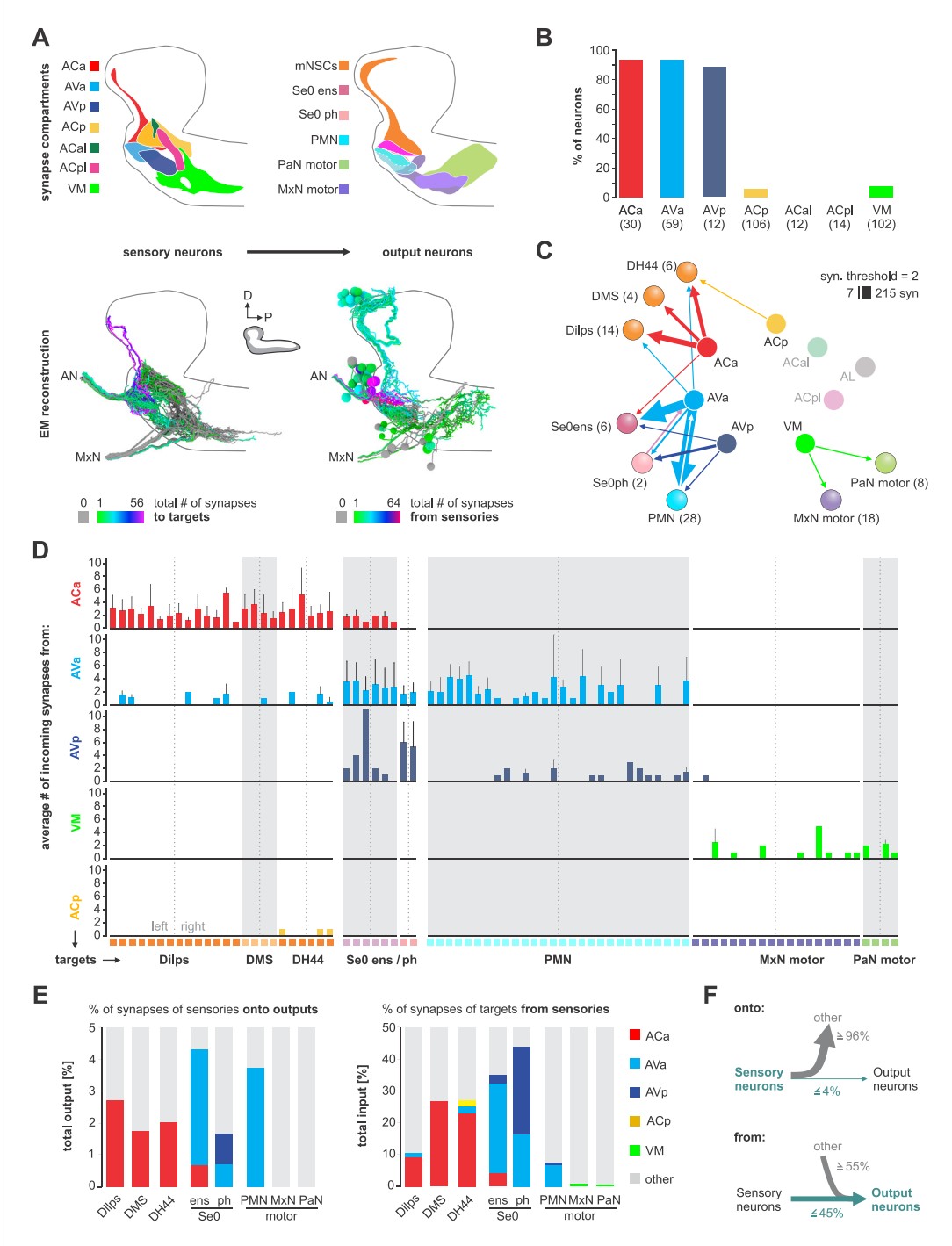

**Figure 3.** Monosynaptic circuits between sensory and output neurons. (**A**) Lateral schematics of the presynaptic sensory compartments and postsynaptic terminals for each output neuron type (upper panel). EM reconstruction of respective neurons (lower panel). Left: sensory neurons are color-coded based on total number of synapses to their monosynaptic targets. Right: output neurons are color-coded based on total number of synapses from sensory neurons. Lateral views show neurons of the right side. (**B**) Percentage of sensory neurons of the respective sensory compartment forming monosynaptic circuits. About 90% of all sensory neurons of ACa, AVa, AVp are part of monosynaptic circuits; in contrast, ACp, ACal, ACpl and VM show little to none. (**C**) Connectivity diagram of axo-dendritic connections between sensory and output neurons. Circles represent previously defined sensory and output compartments (*Figure 2*). Sensory compartments with no monosynaptic reflex connections are faded. (**D**) Connectivity between sensory neurons of different compartments (ACa, AVa, AVp, ACp and VM) and postsynaptic output targets. Each column represents an output target. Whiskers represent standard deviation. (**E**) Left: percentage of synapses of sensory neurons to monosynaptic targets. Right: percentage of

*Figure 3 continued on next page*

*Figure 3 continued*

synapses of output neurons from sensory neurons. (F) Summarizing representation of monosynaptic input-to-output ratio viewed from the sensory neuron side (top) or from the output neuron side (bottom). See *Figure 3—figure supplements 1–5* for detailed connectivity.

DOI: https://doi.org/10.7554/eLife.40247.015

The following figure supplements are available for figure 3:

**Figure supplement 1.** The elemental feeding circuit.

DOI: https://doi.org/10.7554/eLife.40247.016

**Figure supplement 2.** Connectivity principles of monosynaptic reflex connections.

DOI: https://doi.org/10.7554/eLife.40247.017

**Figure supplement 3.** Sensory-to-mNSC connectivity.

DOI: https://doi.org/10.7554/eLife.40247.018

**Figure supplement 4.** Sensory-to-Se0 connectivity.

DOI: https://doi.org/10.7554/eLife.40247.019

**Figure supplement 5.** Sensory-to-PMN connectivity.

DOI: https://doi.org/10.7554/eLife.40247.020

these have the greatest number of monosynaptic contacts from both the enteric system and the pharyngeal sensory neurons.

## Multisynaptic connections to the mushroom body (MB) associative memory circuits

As a contrast to direct input-to-output connections, we additionally looked at connections to a higher brain center for learning and memory, the mushroom body. To this end, we checked previously described projection neurons to the MB calyx (*Eichler et al., 2017*) for inputs from the sensory neurons identified here. Remarkably, the monosynaptic reflex circuit and the multisynaptic MB projections utilize almost completely different set of sensory synaptic compartments (*Figure 6A*). The three compartments that comprise the vast majority of the monosynaptic circuits (ACa, AVa and AVp) have no outputs onto the MB input projection neurons; rather, three new synaptic compartments are utilized (ACp, ACal and ACpl; *Figure 6A,B*). Aside from the AL (from which over 20% of output synapses of the ORNs target the MB calyx via olfactory projection neurons), the most prominent synaptic compartment is the ACal, from which almost 40% of the synapses output onto thermosensory projection neurons. We also note that around 45% of all incoming synapses onto the gustatory projections neurons derive from ACp (*Figure 6C*, right panel). This is also consistent with the view that the ACp is the primary sensory compartment onto which the external and pharyngeal gustatory sensory organs project (*Colomb et al., 2007*; *Hartenstein et al., 2018*).

## Integration of polysynaptic connections onto monosynaptic circuits

We then asked how the hugin neuropeptide (*Drosophila* neuromedin U homolog) circuit, which relays gustatory information to the protocerebrum (*Schlegel et al., 2016*; *Hückesfeld et al., 2016*), would be positioned with respect to the monosynaptic reflex and multisynaptic MB memory circuits. Based on our earlier studies on mapping sensory inputs onto hugin protocerebrum neurons (huginPC) (*Schlegel et al., 2016*; *Hückesfeld et al., 2016*), we were expecting most inputs from the ACp, which is the primary gustatory sensory compartment (*Colomb et al., 2007*; *Hartenstein et al., 2018*). However, most of the huginPC neurons receive inputs from the sensory compartments ACa and AVa, which are the two major monosynaptic compartments that originate from enteric regions. HuginPC neurons do receive inputs from the external and pharyngeal organs (i.e., through sensory compartment ACp), but to a much smaller degree (*Figure 7—figure supplement 1*). Thus, unlike the MB circuit that utilizes a completely new set of sensory inputs, the huginPC circuit is associated with a feeding related monosynaptic circuit.

Based on these observations from the hugin neuropeptide circuit in interconnecting sensory and neuroendocrine outputs, we asked a broader question concerning input-output connections: for any given pair of neurons comprising the monosynaptic reflex circuit, how many additional polysynaptic paths exist and what could be the functional significance of such parallel pathways (*Leonardo, 2005*)? To illustrate, we selected a target neuron from a cluster of neurosecretory and serotonergic modulatory output cells (Dilps and Se0ens), and listed all sensory neurons that make

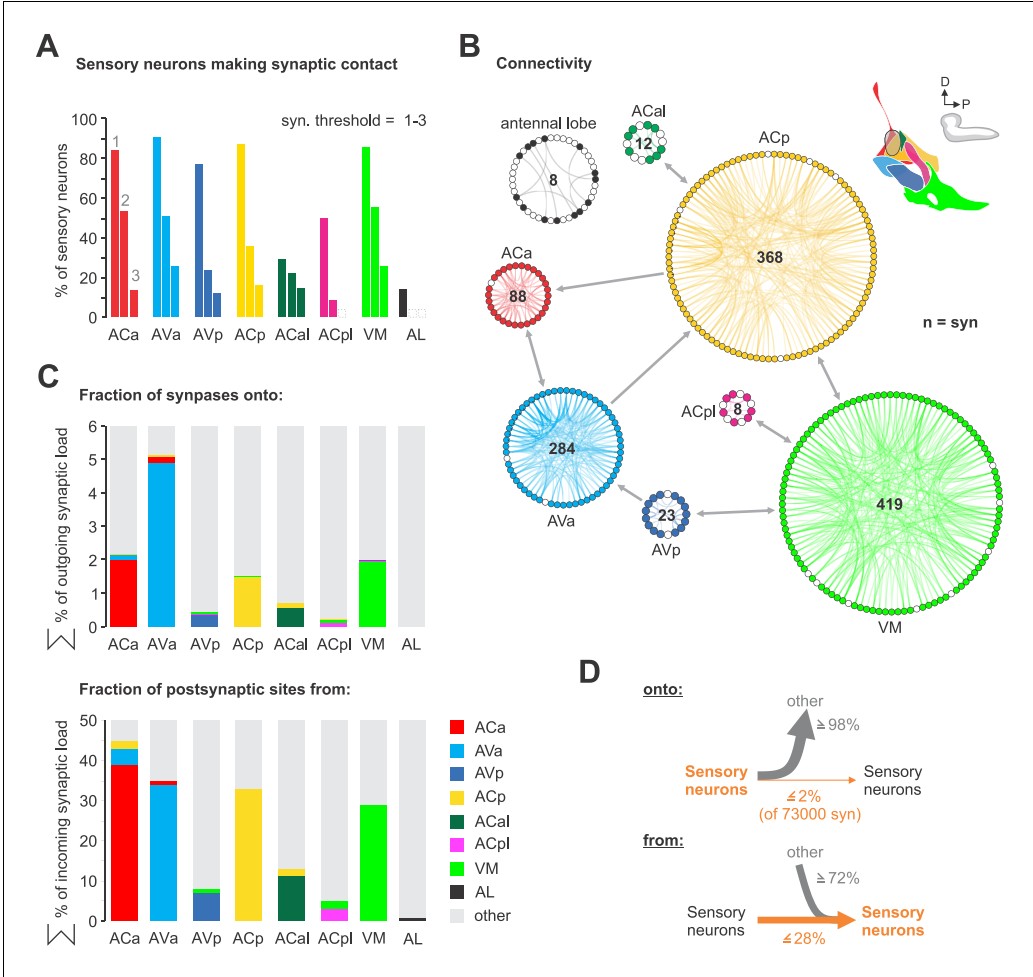

**Figure 4.** Sensory-sensory communication. (**A**) Percentage of sensory neurons of sensory compartments involved in intra compartment sensory connections. Around 80% of all sensory neurons in ACa, AVa, AVp, ACp and VM form intra sensory connections. ACal and ACpl have the lowest number of neurons and also show low number of intra sensory contacts. (**B**) Connectivity diagram of axo-axonic connections between sensory neurons. The circular wheel arrangements represent previously defined sensory compartments (see *Figure 2*). Each small circle within a wheel represents a single neuron. Gray links show inter-cluster connections (max. 10 synapses in one direction). Note that sensory to sensory contacts are made mainly between sensory neurons of the same class, not between classes. (**C**) Percentage of synapses of sensory neurons from and onto sensory neurons. (**D**) Summarizing representation of axo-axonic contact input-to-output ratio viewed from the presynaptic neuron side (top) or from the postsynaptic neuron side (bottom). See *Figure 4—figure supplements 1–2* for detailed connectivity.
DOI: https://doi.org/10.7554/eLife.40247.021

The following figure supplements are available for figure 4:

**Figure supplement 1.** Connectivity principles of inter-sensory connections.
DOI: https://doi.org/10.7554/eLife.40247.022

**Figure supplement 2.** Map of inter-sensory connections.
DOI: https://doi.org/10.7554/eLife.40247.023

monosynaptic connections with at least two synapses (*Figure 7A,B*). We then asked, using the same threshold, how many different di-synaptic paths (2-hop) exist and how often a particular interneuron is used for the different possible converging paths ('degree' of convergence). We also calculated the relative synaptic strengths of the connection among the various paths ('ranking index' of 1.0 represents highest synaptic strength from all possible inputs to the output neuron). Several properties are revealed: (1) different sensory neurons make monosynaptic contacts to a common output target (2) each output neuron can be reached from a given sensory neuron by multiple routes through the use

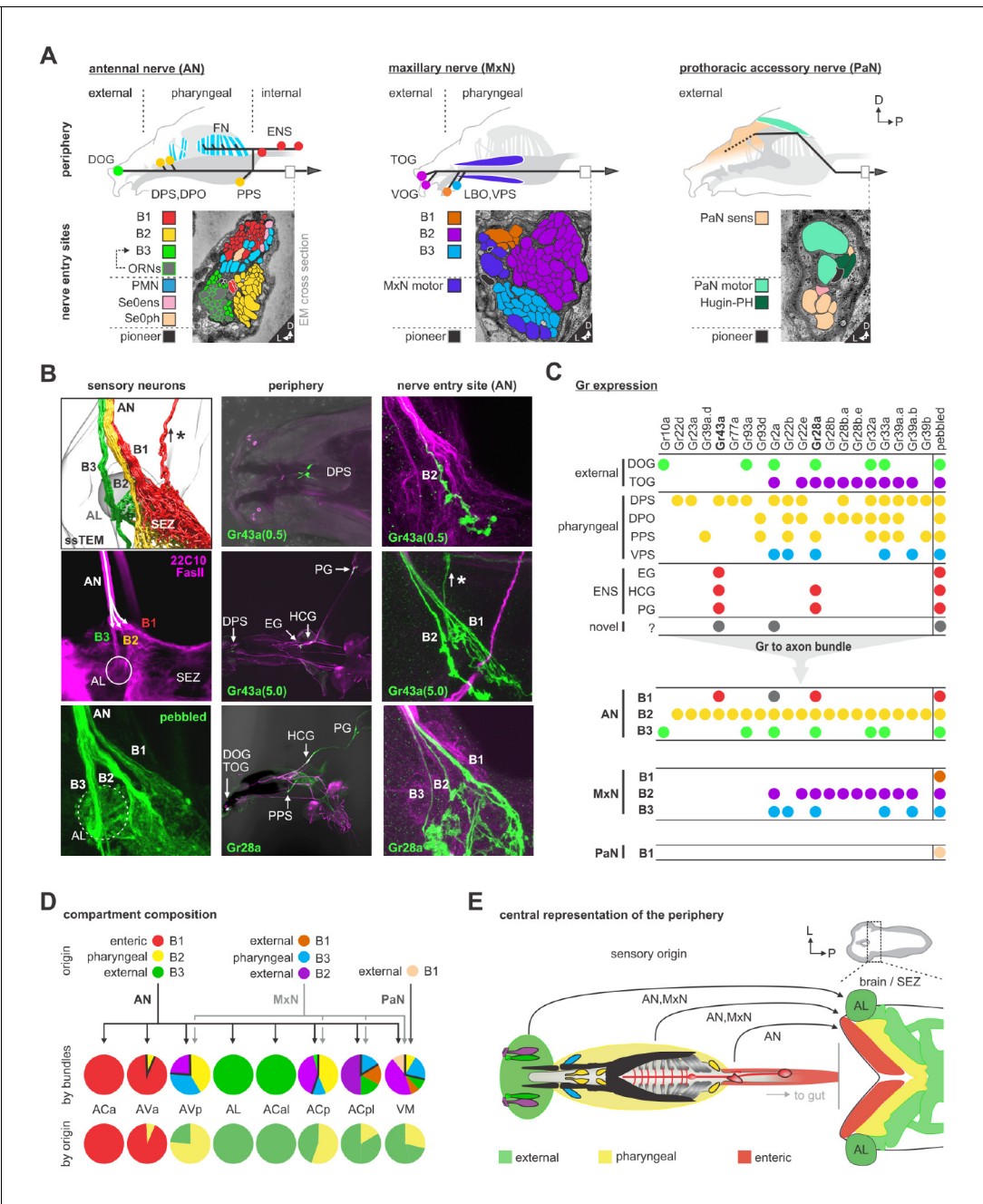

**Figure 5.** Mapping peripheral origin of sensory neurons. (A) Origins and targets of feeding related sensory and motor neurons. The AN comprises motor axons innervating the cibarial dilator muscles (blue striped region) and sensory axons from the dorsal organ ganglion (DOG), pharyngeal sensilla (DPS, DPO, PPS), frontal nerve (FN) and enteric nervous system (ENS). The MxN comprises motor axons innervating the mouth hook elevator and depressor (in purple), labial retractor and salivary gland ductus opener; and sensory axons from the terminal organ ganglion (TOG), ventral organ ganglion (VOG), labial organ (LBO) and pharyngeal sensilla (VPS). The PaN comprises motor axons innervating the dorsal protractor (in green), and sensory neurons with an hypothesized origin in the anterior pharyngeal region (in beige). EM cross section of the right AN, MxN and PaN at nerve entry site (lower panels). Neuronal profiles of all neurons are colored based on their morphological class and origin. (B) Mapping of Gr28a and Gr43a gustatory receptor neuron projection through distinct bundles of the AN from the enteric nervous system. Pebbled-Gal4 was used as a pan-sensory neuronal marker to shows expression in all 3 bundles of the AN. Asterisk marks sensory projections into the protocerebrum. (C) Summary table of selected Gr expression patterns from the peripheral origin (sensory organs and ganglia), and their expression in respective nerve entry site. Note that Gr28a and Gr43a show expression in the ENS (EG, esophageal ganglion; HCG, hypocerebral ganglion; PG, proventricular ganglion), which results in projections through bundle 1 (B1). (D) Sensory compartment composition by peripheral origin. ACa, ACal and AL each derive from a single sensory origin. In contrast, AVa, AVp, ACp, ACpl and VM integrate several sensory origins. Percentage compartment composition is shown by nerve bundles and by origin (enteric, pharyngeal, external). (E) Somatotopic arrangement of sensory axon in the brain and SEZ, showing a layered arrangement that

*Figure 5 continued on next page*

*Figure 5 continued*

mirrors the antero-posterior layout of innervated body structures. The internal layer (red) represents the enteric system. See *Figure 5—figure supplements 1–9* for detailed gustatory receptor expression.

DOI: https://doi.org/10.7554/eLife.40247.024

The following figure supplements are available for figure 5:

**Figure supplement 1.** Gustatory receptor expression.
DOI: https://doi.org/10.7554/eLife.40247.025

**Figure supplement 2.** Gustatory receptor expression.
DOI: https://doi.org/10.7554/eLife.40247.026

**Figure supplement 3.** Gustatory receptor expression.
DOI: https://doi.org/10.7554/eLife.40247.027

**Figure supplement 4.** Gustatory receptor expression.
DOI: https://doi.org/10.7554/eLife.40247.028

**Figure supplement 5.** Gustatory receptor expression.
DOI: https://doi.org/10.7554/eLife.40247.029

**Figure supplement 6.** Gustatory receptor expression.
DOI: https://doi.org/10.7554/eLife.40247.030

**Figure supplement 7.** Gustatory receptor expression.
DOI: https://doi.org/10.7554/eLife.40247.031

**Figure supplement 8.** Gustatory receptor expression.
DOI: https://doi.org/10.7554/eLife.40247.032

**Figure supplement 9.** Gustatory receptor expression.
DOI: https://doi.org/10.7554/eLife.40247.033

of different interneurons (3) a given interneuron can receive inputs from different sensory neurons to target the same output neuron; this would fit the definition of the 'common path' that Sherrington described (*Sherrington, 1906*). These observations hold true for the majority of monosynaptic sensory-output pairs we have examined. However, we have not seen a correlation between the relative synaptic strength and the commonness of the respective paths (i.e., how often a path is used) (*Figure 7—figure supplement 3*).

A potential functional consequence of such circuit architecture can be seen if we now include all the sensory inputs onto the interneurons. As an example, we take Dilp 1L as the common output, and the interneurons H1 (a huginPC neuron) and 'S' (not previously described) as two of the polysynaptic paths onto a common output (*Figure 7C*, layer 2). One consequence of such superimposition is that the amount of sensory information that can reach a common output neuron can be significantly increased. In this case, the Dilp 1L neuron, in addition to receiving inputs from four sensory neurons (from monosynaptic paths, layer 1), now receives inputs from seven new sensory neurons through the interneuron 'S', and eight new sensory neurons from interneuron H1. Furthermore, these additional sensory neurons derive from new peripheral regions (e.g., pharyngeal in addition to enteric). Note also that the two interneurons themselves interact, thus increasing the number of paths available between any sensory-output pair (also see *Figure 7—figure supplement 2*). The interneurons could also sharpen sensory information by inhibiting parallel pathways, for example through feed-forward inhibition.

Another circuit layer comes into play when tri-synaptic (3-hop) paths are analyzed (*Figure 7C*, layer 3). For example, the interneuron 'Ag' (not previously described) brings in a different set of new sensory inputs that converge onto a common target (Dilp 1L). In addition, it receives sensory inputs from layer 2 sensory neurons. We thus observe a circuit architecture where new interneurons that target the same output neurons are successively layered, and which receive sensory input from the previous layer as well as from completely new set of inputs. Extending this to the full set of feeding output neurons (mNSCs, Se0ens, Se0ph and PMNs) illustrates the increase in the sensory neuron number and in the peripheral origin that can be gained by integrating polysynaptic connections onto monosynaptic reflex circuits (*Figure 7D*; *Figure 7—figure supplements 2* and *3*): For example, monosynaptic paths to the mNSCs would only allow sensory inputs from enteric origin, whereas polysynaptic paths would allow sensory inputs from enteric, pharyngeal and external origins (*Figure 7D*, top panel).

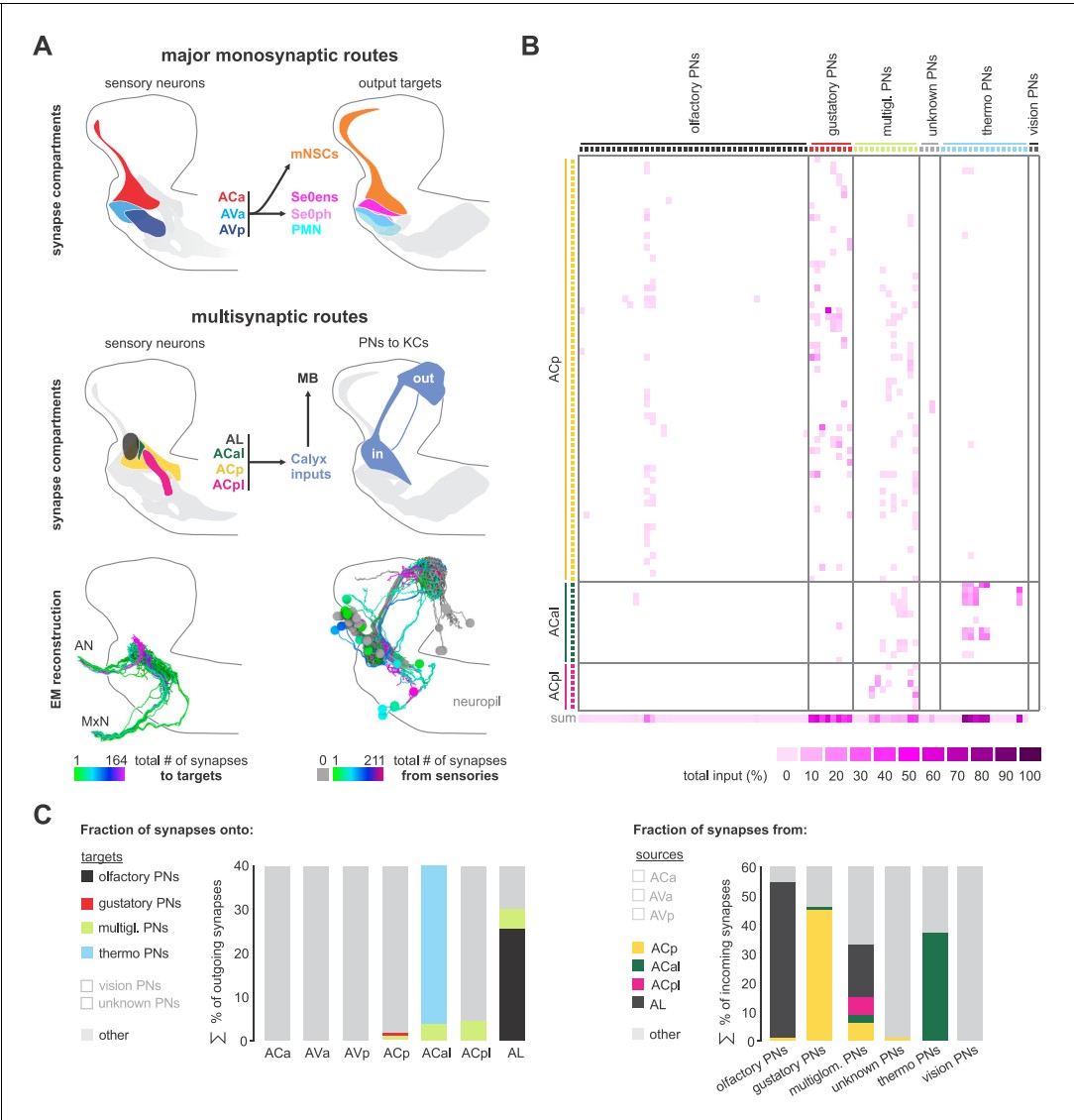

**Figure 6.** Multisynaptic sensory inputs onto mushroom body circuits. (A) Schematic of major monosynaptic routes (top panel). Connectivity between presynaptic sensory neurons (ACa, AVa and AVp) and postsynaptic outputs (mNSCs, Se0ens, Se0ph and PMNs). Schematic of multisynaptic routes to the mushroom body (middle panel). Connectivity between presynaptic sensory neurons (antennal lobe, ACal, ACp, ACpl) and postsynaptic projection neurons to the calyx (PNs to KCs). EM reconstruction of respective neurons (lower panel); Left: sensory neurons (olfactory receptor neurons are excluded) are color-coded based on total number of synapses to the projection neurons. Right: projection neurons are color-coded based on total number of synapses from sensory neurons. Lateral views show neurons of the right brain hemisphere. (B) Adjacency matrix showing sensory-to-PN connectivity, color-coded by percentage of inputs on PN dendrites. Or35a-PN is essentially the only olfactory projection neuron that receives multisensory input from non-olfactory receptor neurons of the ACp (primary gustatory center). (C) Left: percentage of presynapses of sensory neurons to PNs. Right: percentage of postsynapses of PNs from sensory neurons.
DOI: https://doi.org/10.7554/eLife.40247.034

Finally, we asked whether any output neurons of the mushroom body (MBONs, *Eichler et al., 2017*) connect to any of the interneurons that comprise the different layers of the feeding circuit. In other words, do the MBONs target the interneurons, that in turn target the outputs (*Figure 7E*; *Figure 7—figure supplement 4*). Strikingly, the MBON-f1 makes monosynaptic connections to a large number of interneurons that target all classes of feeding output neurons (*Figure 7F* shows example for mNSCs; see *Figure 7—figure supplement 5* for Se0ens, Se0ph and PMN). Furthermore, this MB module (consisting of ORNs, projection neurons to the Kenyon cells and MBONs) can be placed on top of the existing feeding circuit, since the interneurons targeted by MBON-f1 are shared by those

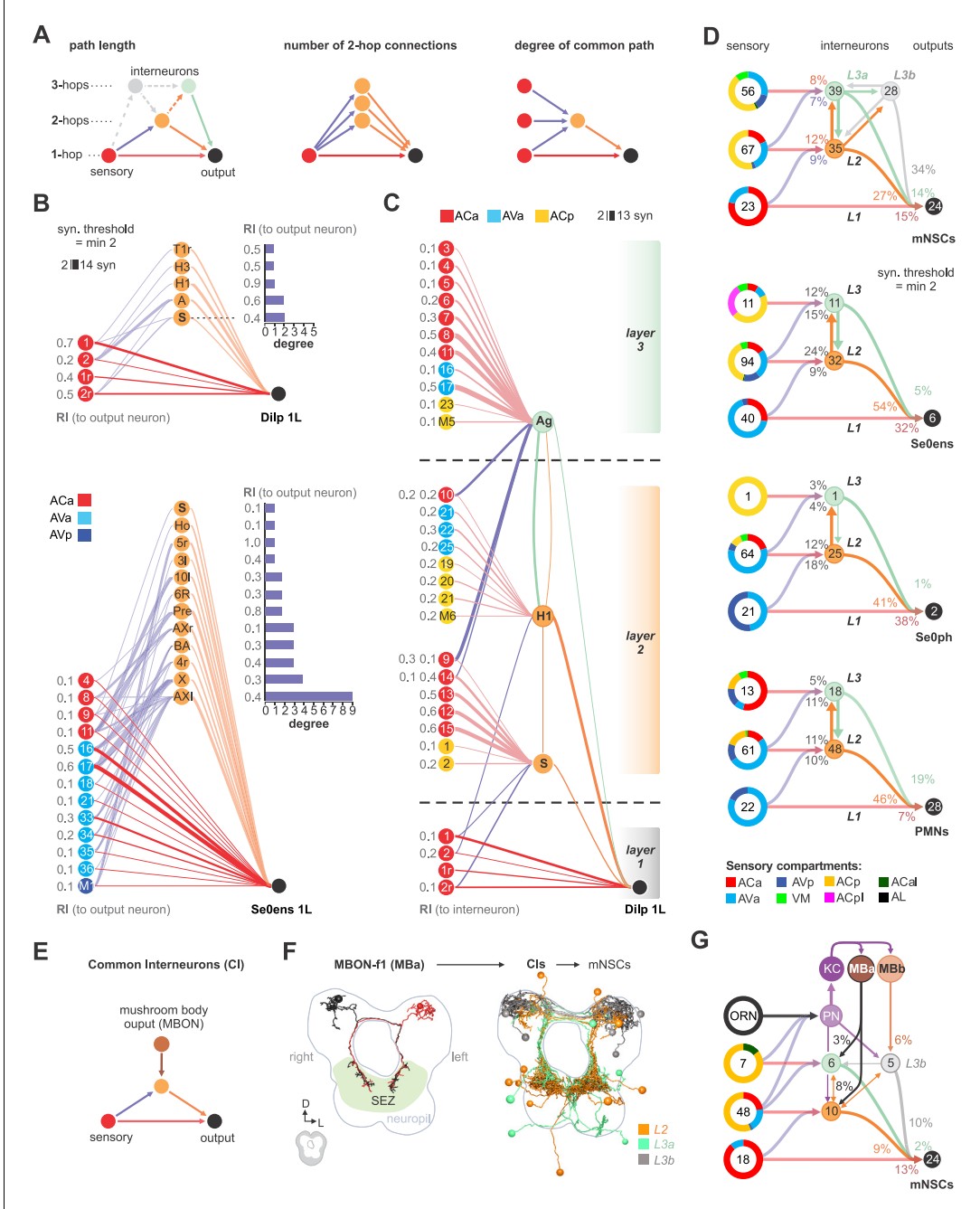

**Figure 7.** Integration of polysynaptic connections onto monosynaptic circuits. (**A**) Left panel: Illustration of direct (1-hop) sensory to output neuron connections and indirect (2-hop, 3-hop) paths which involve 1 or 2 interneurons to reach the same output neuron. 3-hop connections through interneurons which are not part of the direct upstream of the output neurons were not considered. Middle panel: Illustration of sensory divergence, which defines the number of possible paths to reach the same target neuron through different interneurons. Right panel: Illustration of sensory convergence, which defines how often (degree) a common path is used by different sensory neurons to reach the same output neuron. (**B**) Upper panel: All 1- and 2-hop connections for one cell of the Dilp cluster of neurosecretory output cells (Dilps, DMS and DH44), using a synaptic threshold of 2. Lower panel: All 1- and 2-hop connections for one cell of the Se0 cluster, using a synaptic threshold of 2. Ranking index (**RI**) shows the relative synaptic strength of every connection compared to the total synaptic input of the output neuron (1.0 represents the highest from all possible inputs to the output neuron). (**C**) Superimposition of selected 2- and 3-hop paths. Layer 1 shows a basic sensory-to-Dilp circuit. Layer 2 represents alternative paths through directly (2-hop) connected interneurons ('S' and 'H1/HuginPC left 1') and their sensory inputs, using a synaptic threshold of 2. Both interneurons integrate a completely different set of sensory neurons from different sensory compartments to connect onto the basic sensory-to-Dilp circuit. Layer 3 represents paths through indirectly connected interneurons ('Ag') and their sensory inputs, using a synaptic threshold of 2. This interneuron receives sensory information from layer 2 and also integrates a completely different set of sensory neurons onto the basic sensory-to-Dilp

*Figure 7 continued on next page*

*Figure 7 continued*

circuit. Ranking index (RI) shows the relative synaptic strength of every sensory-to-interneuron connection compared to the total synaptic input of the interneurons. (D) Summarizing representation of all monosynaptic sensory-to-output connections (grouped targets: mNSCs, Se0ens, Se0ph and PMNs), and their alternative paths through interneurons to reach one cell of the target group, using a synaptic threshold of 2. Note that nearly all alternative paths (interneurons) of layer 2 and 3 (L2, L3) receive monosynaptic input from other sensory neurons (synaptic threshold = 2), thus integrating a completely different set of sensory neurons onto the basic reflex circuits. For mNSCs: Layer 3 is divided into interneurons which receive monosynaptic sensory inputs (L3a) and those which do not receive monosynaptic input from any sensory neuron (L3b), *Figure 7—figure supplement 2B*. Percentages represent fraction of synapses from upstream neurons (arrows). Numbers within circles represent number of neurons. Percentage sensory composition (the three left donut circles) is shown by sensory compartment. See *Figure 7—figure supplements 1–3* for detailed connectivity and path numbers. (E) Illustration of direct (1-hop) sensory to output neuron connections and indirect (2-hop) paths which connect mushroom body output neurons (MBONs) to reach the same output neuron, thus representing final common interneurons (CI) (F) Left: EM reconstruction of MBON-f1 (part of MBa, *Figure 7—figure supplement 4A,D*) which is the only MBON with projections to the SEZ. Right: EM reconstruction of all presynaptic interneurons to the mNSCs that also receive monosynaptic contacts from MBONs (synaptic threshold = 2). Interneurons are color coded based on mNSC path layers (L2, L3a, L3b) (G) Summarizing representation of all common interneurons to mNSCs that receive monosynaptic input from MBONs. MBONs are divided into two groups. MBa, which synapses onto interneurons (L3a) that receive monosynaptic contacts from sensory neurons and MBb, which synapses onto interneurons (L3b) that do not integrate sensory information onto the basic sensory-to-mNSC circuit. Note that projection neurons to the mushroom body calyx (PN, KC, *Figure 7*) also act as part of 3-hop paths from layer 2 and layer 3 sensory neurons to the mNSCs. Percentages represent fraction of synapses from MBONs to CIs and CIs to mNSCs. Numbers within circles represent number of neurons. Percentage sensory composition (the four left donut circles) is shown by sensory compartment. Exclusive sensory-to-output connections are not shown See *Figure 7—figure supplements 4–5* for detailed connectivity and neuroanatomy.

DOI: https://doi.org/10.7554/eLife.40247.035

The following figure supplements are available for figure 7:

**Figure supplement 1.** Connectivity between presynaptic sensory neurons of mNSCs and huginPC neurons.
DOI: https://doi.org/10.7554/eLife.40247.036
**Figure supplement 2.** Connectivity of interneurons within the feeding circuits.
DOI: https://doi.org/10.7554/eLife.40247.037
**Figure supplement 3.** Quantification of alternative paths onto output neurons.
DOI: https://doi.org/10.7554/eLife.40247.038
**Figure supplement 4.** Connectivity of mushroom body output neurons onto the feeding circuits.
DOI: https://doi.org/10.7554/eLife.40247.039
**Figure supplement 5.** EM reconstruction of common interneurons.
DOI: https://doi.org/10.7554/eLife.40247.040

that comprise the previous layers of the feeding circuit (*Figure 7G*; *Figure 7—figure supplement 4*). This MBON may thus be representative of a 'psychomotor' neuron described by Cajal (*Ramón y Cajal, 1894*; see also *Swanson, 2011*), which acts higher and in parallel to a reflex circuit.

In summary, we propose that the different path possibilities allow different strength and combination of sensory inputs to be evaluated, which would then determine which synaptic path will dominate to a given output. Such multisensory integration via multiple parallel pathways would be necessary to make sense of a complex, multimodal world, and to better choose a behavioral response.

## Discussion

We provide a comprehensive synaptic map of the sensory and output neurons that underlie food intake and metabolic homeostasis in *Drosophila* larva. Seven topographically distinct sensory compartments, based on modality and peripheral origin, subdivide the SEZ, a region with functional similarities to the vertebrate brainstem. Sensory neurons that form monosynaptic connections are mostly of enteric origin, and are distinct from those that form multisynaptic connections to the mushroom body (MB) memory circuit. Different polysynaptic connections are superimposed on the monosynaptic input-ouput pairs that comprise the reflex arc. Such circuit architecture may be used for controlling feeding reflexes and other instinctive behaviors.

### Elemental circuit for feeding

Reflex circuits represent a basic circuit architecture of the nervous system, whose anatomical and physiological foundations were laid down by Cajal and Sherrington (*Ramón y Cajal, 1894*;

*Sherrington, 1906*; *Swanson, 2011*).The *Drosophila* larval feeding reflex circuit comprises the motor neurons that innervate the muscles involved in pharyngeal pumping, as well as the neurosecretory neurons that target the endocrine organs. They also include a cluster of serotonergic neurons that innervate the entire enteric nervous system, and which may have neuromodulatory effects on the feeding system in a global manner. The vast majority of output neurons are targeted monosynaptically from a set of topographically distinct sensory synaptic compartments in the CNS. These compartments target the output neurons in overlapping domains: the first, ACa, targets all neuroendocrine cells as well as the serotonergic neurons; the second, AVa, targets a subset of neuroendocrine cells, the serotonergic neurons and most of the pharyngeal motor neurons, while the third, AVp, targets the serotonergic neurons and a different set of pharyngeal motor neurons. With these outputs, one can in principle fulfill the most basic physiological and behavioral needs for feeding: neurosecretory cells for metabolic regulation and pharyngeal motor neurons for food intake. This set of monosynaptic connections can thus be seen to represent an elemental circuit for feeding, since the connections between the input and output neurons cannot be broken down any further.

Vast majority of the sensory inputs comprising this 'elemental feeding circuit' derive from the enteric nervous system to target the pharyngeal muscles involved in food intake and neuroendocrine output organs. However, there is a small number of monosynaptic reflex connections that originate from the somatosensory compartment. The output neurons targeted by these somatosensory neurons are motor neurons that control mouth hook movements and head tilting, movements which are involved in both feeding and locomotion. In this context, it is noteworthy that monosynaptic reflex connections are found to a much lesser degree in the larval ventral nerve cord, which generates locomotion (unpublished data from *Ohyama et al., 2015*). An analogous situation exists in *C. elegans*, where majority of the monosynaptic reflex circuits are found in the head motor neurons and not in the body (*Yan et al., 2017*). One reason could be due to the relative complexity in the response necessary for food intake as compared to locomotion. For example, a decision to finally not to swallow a harmful substance, once in the mouth, may require a more local response, for example muscles limited to a very specific region of the pharynx and esophagus, where monosynaptic arc might suffice. By contrast, initiating escape behaviors requires a more global response with respect to the range and coordination of body movements involved, although it also employs multimodal sensory integration via a multilayered circuit (*Ohyama et al., 2015*).

## Monosynaptic connections between the sensory neurons

The inter-sensory connections show a combination of hierarchical and reciprocal connections, which may increase the regulatory capability and could be especially important for monosynaptic circuits. By contrast, very few monosynaptic connections exist between the larval olfactory, chordotonal or nociceptive class IV sensory neurons in the body (*Ohyama et al., 2015*; *Jovanic et al., 2016*; *Gerhard et al., 2017*). Interestingly, there is also a much higher percentage of intersensory connections between olfactory receptor neurons in the adult as compared to the larva, which could function in gain modulation at low signal intensities (*Tobin et al., 2017*). This might be attributable to adults requiring faster processing of olfactory information during flight navigation (or mating), and/or to minimize metabolic cost (*Wilson, 2014*). Whether such explanation also applies to the differences in intersensory connection between the different types of sensory neurons in the larvae remains to be determined.

## Superimposition of polysynaptic pathways onto monosynaptic circuits

We found very few cases where a monosynaptic path between any sensory-output pair is not additionally connected via a polysynaptic path. An interesting question in the context of action selection mechanism is which path a sensory signal uses to reach a specific target neuron. For example, a very strong sensory signal may result in a monosynaptic reflex path being used. However, a weaker sensory signal may result in using a different path, such as one with less threshold for activation. This would also enable the integration of different types of sensory signals through the usage of multiple interneurons, since the interneurons may receive sensory inputs that are not present in monosynaptic connections. For example, sensory neurons can target the neuroendocrine cells directly (monosynaptically), or through a hugin interneuron (di-synaptically). The sensory compartments that directly target the neuroendocrine cells are of enteric origin; however, when hugin neurons are utilized as

interneurons, not only is the number of sensory neurons from the same sensory compartment increased, but sensory neurons are added from a completely new peripheral origin. Thus, the hugin interneurons enable sensory inputs from different peripheral origins, for example to integrate enteric inputs with pharyngeal gustatory inputs, to influence an output response, which, in this case, is to stop feeding (*Schoofs et al., 2014a*).

The coexistence of polysynaptic and monosynaptic paths could also be relevant for circuit variability and compensation (*Leonardo, 2005*; *Marder and Goaillard, 2006*): destruction of any given path would still enable the circuit to function, but with more restrictions on the precise types of sensory information it can respond to. In certain cases, this may even lead to strengthening of alternate paths as a form of synaptic plasticity.

An open issue is how the sensory synaptic compartments might be connected to the feeding central pattern generators (CPGs) which have been demonstrated to exist in the SEZ (*Schoofs et al., 2010*; *Hückesfeld et al., 2015*), especially since CPGs are defined as neural circuits that can generate rhythmic motor patterns in the absence of sensory input. However, the modulation of CPG rhythmic activity can be brought about by sensory and neuromodulatory inputs (*Marder and Bucher, 2001*; *Marder, 2012*). A complete circuit reconstruction of the larval SEZ circuit may shed some light on the circuit structure of feeding CPGs.

## Multisynaptic sensory inputs onto memory circuits

A more complex circuit architecture is represented by the MB, the site of associative learning and memory in insects: a completely different set of sensory synaptic compartments is used to connect the various projection neurons to the MB calyx. Thus, the MB module is not superimposed onto the monosynaptic reflex circuits but rather forms a separate unit. The classical studies by Pavlov demonstrated conditioned reflex based on an external signal and an autonomic secretory response in response to food (*Pavlov, 2010*; *Todes, 2001*). Although a comparable autonomic response has not been analyzed in the larvae, analogous associative behavior based on odor choice response has been well studied (*Aceves-Piña and Quinn, 1979*; *Gerber and Stocker, 2007*; *Eichler et al., 2017*; *Widmann et al., 2018*). It is also noteworthy that in the *Aplysia*, classical conditioning of the gill withdrawal reflex involves monosynaptic connections between a sensory neuron (mechanosensory) and a motor neuron, and neuromodulation by serotonin (*Bailey et al., 2000*). This constellation has similarities with the elemental feeding circuit consisting of sensory, motor and serotonergic modulatory neurons. For more complex circuits of feeding behavior in the mouse, a memory device for physiological state, such as hunger, has been reported involving synaptic and neuropeptide hormone circuits (*Yang et al., 2011*). Functional studies on MB output neurons such as the MBON-f1, which may be part of a 'psychomotor' pathway (*Ramón y Cajal, 1894*) and which targets a number of interneurons that connect to the neurosecretory, serotonergic and pharyngeal motor neurons, may help address how memory circuits interact with feeding circuits.

## Control of reflexes

Feeding behavior manifests itself from the most primitive instincts of lower animals, to deep psychological and social aspects in humans. It encompasses cogitating on the finest aspects of food taste and the memories evoked by the experience, to sudden reflex reactions upon unexpectedly biting down on a hard seed or shell. Both of these extremes are mediated, to a large degree, by a common set of feeding organs, but the way these organs become utilized can vary greatly. The architecture of the feeding circuit described here allows the various types of sensory inputs to converge on a limited number of output responses. The monosynaptic pathways would be used when fastest response is needed. The presence of polysynaptic paths would enable slower and finer control of these reflex events by allowing different sensory inputs, strengths or modalities to act on the monosynaptic circuit. This can be placed in the context in the control of emotions and survival circuits (*LeDoux, 2012*), or by cortex regulation of basic physiological or autonomic processes (*Dum et al., 2016*). In a striking example, pupil dilation, a reflex response, has been used as an indicator of cognitive activity (*Hess and Polt, 1964*; *Kahneman and Beatty, 1966*; *Larsen and Waters, 2018*). Here, a major function of having more complex circuit modules on top of monosynaptic circuits may be to allow a finer regulation of feeding reflexes, and perhaps of other reflexes or instinctive behaviors.

As an outlook, our analysis provides an architectural framework of how a feeding circuit is organized in the CNS. The circuit is divided into two main axes that connect the input to the output systems: the sensory-neurosecretory cell axis and the sensory-motor neuron axis (*Swanson, 2011*). The sensory system targets overlapping domains of the output neurons; for example, a set of sensory neurons targets exclusively the neuroendocrine cells, other targets both neuroendocrine and pharyngeal motor neurons, and another just the pharyngeal motor neurons. The inputs derive mostly from the internal organs. These connections form the monosynaptic reflex circuits. With these circuits, one can perform the major requirements of feeding regulation, from food intake and ingestion to metabolic homeostasis. Additional multisynaptic circuits, such as the CPGs, those involving sensory signaling from the somatosensory system (external inputs), or those comprising the memory circuits, are integrated or added to expand the behavioral repertoire of the animal (*Figure 8*). Although circuit construction may proceed from internal to the external, the sequence is reversed in a feeding animal: the first sensory cues are external (olfactory), resulting in locomotion (somatic muscles) that can be influenced by memory of previous experience; this is followed by external taste cues, resulting in food intake into the mouth; the final action is the swallowing of food, involving pharyngeal and enteric signals and reflex circuits. However, regardless of the types of sensory inputs, and whether these are transmitted through a reflex arc, a memory circuit or some other multisynaptic circuits in the brain, they will likely converge onto a certain set of output neurons, what Sherrington referred to as the 'final common path' (*Sherrington, 1906*). The current work is a first step towards finding the common paths.

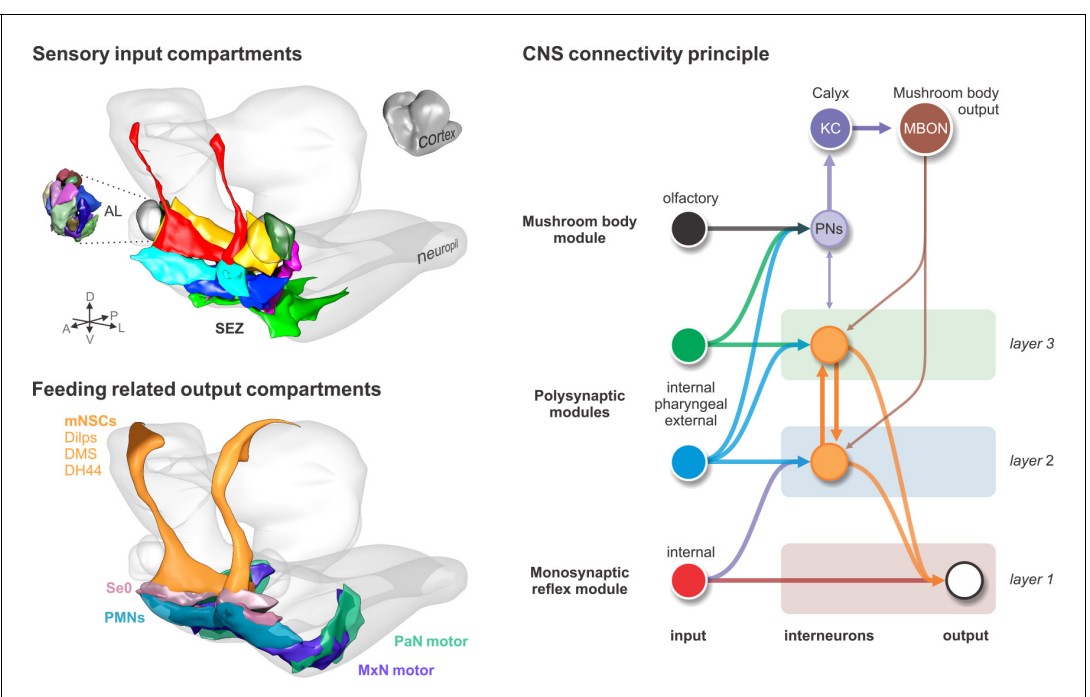

**Figure 8.** Input-output synaptic organization of the larval feeding system and its connectivity architecture in the brain. Sensory input compartments: Glomerular compartmentalization of the larval antennal lobe (AL) compared to glomerular-like compartmentalization of the subesophageal zone (SEZ). Non-overlapping digital 3D model delineates compartments based on synapse similarity score. Feeding related output compartments: 3D model summarizes the synaptic compartments of median neurosecretory cells (mNSCs), modulatory serotonergic output neurons (Se0) and feeding motor neurons (PMNs, MxN motor and PaN motor). CNS connectivity principle: Different polysynaptic modules are integrated onto existing monosynaptic circuits, or added separately as new multisynaptic circuits, for example the mushroom body module.
DOI: https://doi.org/10.7554/eLife.40247.041

## Materials and methods

### Neuronal reconstruction

All reconstructions were based on an ssTEM (serial section transmission electron microscope) data set of a complete nervous system of a 6-h-old [iso] CantonS G1 x w1118 larva as described in (*Ohyama et al., 2015*). Using a modified version of the web-based software CATMAID (*Saalfeld et al., 2009*) we manually reconstructed neurons' skeletons and annotated synapses following the methods described in (*Ohyama et al., 2015*) and (*Schneider-Mizell et al., 2016*). Sensory and motor neurons were identified by reconstructing all axons in the antennal nerve, maxillary nerve and the prothoracic accessory nerve. Further, neurons with their soma in the brain and projections through one of the three pharyngeal nerves have been identified as motor neurons and serotonergic output neurons. All annotated synapses represent fast, chemical synapses equivalent to previously described typical criteria: thick black active zones, pre- and postsynaptic membrane specializations (*Prokop and Meinertzhagen, 2006*).

### Morphology similarity score

To neuron morphologies (*Figure 1*; *Figure 1—figure supplements 2–4*; *Figure 2—figure supplement 6*), we used a morphology similarity score described by *Kohl et al. (2013)*. Briefly, reconstructions of neurons are converted to 'dotprops', 3d positions with an associated tangent vector: for each dotprop from a query neuron, the closest point on a target neuron was determined and scored by distance and the absolute dot product of their two tangent vectors. The total similarity score is the average score over all point pairs between query neuron Q and target neuron T:

$$S(Q,T) = \frac{1}{n} \sum_{i=1}^{n} \sqrt{|q_i \cdot t_j| e^{-\frac{d_{ij}^2}{2\sigma^2}}}$$

where n is the number of points in the query neuron, $d\_{ij}$ is the distance between point i in the query neuron and its nearest neighbor, point j, in the target neuron and $q\_i$ and $t\_j$ are the tangent vectors at these points. $\sigma$ determines how close in space points must be to be considered similar. For our calculations, we used $\sigma$ of 2 um. Similarity score algorithm was implemented in a Blender plugin (*Schlegel, 2018*).

### Synapse similarity score

To calculate similarity of synapse placement between two neurons, we calculated the synapse similarity score (*Figure 2*; *Figure 2—figure supplements 1–6*; *Figure 3—figure supplements 3–5*):

$$f(i_s, j_k) = e^{\frac{-d_{sk}^2}{2\sigma^2}} e^{\frac{|n(i_s) - n(j_k)|}{n(i_s) + n(j_k)}}$$

With the overall synapse overall synapse similarity score for neurons i and j being the average of $f(i\_s, j\_k)$ over all synapses s of i. Synapse k being the closest synapse of neuron j to synapse s [same sign (pre-/post-synapse) only]. $d\_{sk}$ being the linear distance between synapses s and k. Variable $\sigma$ determines which distance between s and k is considered as close. $n(j\_k)$ and $n(i\_s)$ are defined as the number of synapses of neuron j/i that are within a radius $\omega$ of synapse k and s, respectively (same sign only). This ensures that in case of a strong disparity between $n(i\_s)$ and $n(j\_k)$, $f(i\_s, j\_k)$ will be close to zero even if distance $d\_{sk}$ is very small. Values used: $\sigma = \omega = 2000$ nm. Similarity score algorithm was implemented in a Blender plugin (*Schlegel, 2018*).

### Normalized connectivity similarity score

To compare connectivity between neurons (*Figure 2—figure supplement 6*), we used a modified version of the similarity score described by *Jarrell et al. (2012)*:

$$f(A_{ik}, A_{jk}) = \min(A_{ik}, A_{jk}) - C_1 \max(A_{ik}, A_{jk}) e^{-C_2 \min(A_{ik}, A_{jk})}$$

With the overall connectivity similarity score for vertices i and j in adjacency matrix A being the sum of $f(A\_{ik}, A\_{jk})$ over all connected partners k. $C\_1$ and $C\_2$ are variables that determine how similar two vertices have to be and how negatively a dissimilarity is punished. Values used were: $C\_1 = 0.5$

and $C_2 = 1$. To simplify graphical representation, we normalized the overall similarity score to the minimal (sum of $-C_1 \max(A_{ik}, A_{jk})$ over all k) and maximal (sum of $\max(A_{ik}, A_{jk})$ over all k) achievable values, so that the similarity score remained between 0 and 1. Self-connections $(A_{ii}, A_{jj})$ and $A_{ij}$ connections were ignored.

## Clustering

Clusters for dendrograms were created based on the mean distance between elements of each cluster using the average linkage clustering method. Clusters were formed at scores of 0.06 for synapse similarity score (*Figure 2*; *Figure 2—figure supplements 1–5*).

## Percentage of synaptic connections

Percentage of synaptic connections was calculated by counting the number of synapses that constitute between neuron A and a given set of pre- or postsynaptic partners divided by the total number of either incoming our outgoing synaptic connections of neuron A. For presynaptic sites, each postsynaptic neurite counted as a single synaptic connection (*Figure 3*; *Figure 3—figure supplements 3–5*; *Figure 4*; *Figure 6*; *Figure 7*; *Figure 7—figure supplements 1* and *4*).

## Ranking index

Ranking index was calculated by counting the number of synapses that constitute between neuron A and a given target neuron B divided by the highest number of synapses among all incoming synaptic connections of target neuron B (*Figure 7*; *Figure 7—figure supplements 2* and *4*).

## Neuronal representation

Neurons were rendered with Blender 3D (www.blender.org) and edited in Adobe Corel Draw X7 (www.corel.com). A script for a CATMAID-Blender interface is on Github (*Schlegel, 2018*; copy archived at https://github.com/elifesciences-publications/CATMAID-to-Blender).

## Graphs

Graphs were generated using Excel for Mac v15.39 (www.microsoft.com), Sigma Plot 12.0 (www.sigmaplot.com) and edited in Corel Draw X7.

## Flies

The following GAL4 driver and UAS effector lines were used: Gr2a-GAL4 (Bloomington #57589), Gr10a-GAL4 (Bloomington #57597), Gr22b-GAL4 (Bloomington #57604), Gr22e-GAL4 (Bloomington #57608), Gr23a-GAL4 (Bloomington #57611), Gr28a-GAL4 (Bloomington #57614 and #57613), Gr28b-GAL4 (*Scott et al., 2001*), Gr28b.a-GAL4 (Bloomington #57615), Gr28b.e-GAL4 (Bloomington #57621), Gr32a-GAL4 (Bloomington #57622), Gr33a-GAL4 (Bloomington #57624), Gr39a.a-GAL4 (Bloomington #57631), Gr39a.b-GAL4 (Bloomington #57632), Gr39a.d-GAL4 (Bloomington #57634), Gr39b-GAL4 (Bloomington #57635), Gr43a-GAL4 (Bloomington #57636 and #57637), Gr43a$^{GAL4}$ knock-in (*Miyamoto et al., 2012*), Gr57a-GAL4 (Bloomington #57642), Gr58b-GAL4 (Bloomington #57646), Gr59a-GAL4 (Bloomington #57648), Gr59d-GAL4 (Bloomington #57652), Gr63a-GAL4 (Bloomington #57660), Gr66a-GAL4 (*Scott et al., 2001*), Gr68a-GAL4 (Bloomington #57671), Gr77a-GAL4 (Bloomington #57672), Gr93a-GAL4 (Bloomington #57679), Gr93b-GAL4 (Bloomington #57680), Gr93c-GAL4 (Bloomington #57681), Gr93d-GAL4 (Bloomington #57684), 94a-GAL4 (Bloomington #57686), Orco-GAL4 (Bloomington #23909) and 10X-UAS-mCD8::GFP (Bloomington #32184)

## Immunohistochemistry

Dissected larval brains with attached CPS and intact pharyngeal nerves were fixed for 1 hr in paraformaldehyde (4%) in PBS, rinsed with PBS-T and blocked in PBS-T containing 5% normal goat serum. For antibody stainings of Gr-GAL4 >10xUAS-mCD8::GFP primary antibody were conjugated goat anti-GFP (1:500, Abcam, ab6662), mouse anti-fasciclin2 (1:500, DSHB) and mouse anti-22C10 (1:500, DSHB) and the secondary antibody was anti-mouse Alexa Flour 568 (1:500, Invitrogen). Brains were rinsed with PBS-T and mounted in Mowiol (Roth, 0713). For antibody stainings of Gr43a-GAL4 > 10xUAS-mCD8::GFP primary antibody were rabbit anti-GFP (1:500, Abcam, ab6556), mouse

anti-fasciclin2 (1:500, DSHB) and mouse anti-22C10 (1:500, DSHB) and the secondary antibody were anti-rabbit Alexa Flour 488 (1:500, Invitrogen) and anti-mouse Alexa Flour 568 (1:500, Invitrogen). Brains were rinsed with PBS-T and dehydrated through an ethanol-xylene series and mounted in DPX. Imaging was carried out using a Zeiss LSM 780 confocal microscope with a 25x objective (Zeiss).

## Acknowledgements

We thank Sarah Brenner, Benjamin White, Henning Fenselau, Andreas Thum and William Schafer for their timely contributions, Michael Winding for helping clarify the MBON nomenclature and the Bloomington Stock Center for fly strains. We also thank all the 'tracers' for their contribution to the EM reconstruction.

## Additional information

### Funding

| Funder | Author |
|---|---|
| Deutsche Forschungsge-meinschaft | Anton Miroschnikow<br>Philipp Schlegel<br>Andreas Schoofs<br>Sebastian Hueckesfeld<br>Michael J Pankratz |
| Howard Hughes Medical Institute | Feng Li<br>Casey M Schneider-Mizell<br>Richard D Fetter<br>James W Truman<br>Albert Cardona |

The funders had no role in study design, data collection and interpretation, or the decision to submit the work for publication.

### Author contributions

Anton Miroschnikow, Conceptualization, Data curation, Formal analysis, Investigation, Visualization, Methodology, Writing—original draft, Writing—review and editing; Philipp Schlegel, Conceptualization, Data curation, Formal analysis, Methodology; Andreas Schoofs, Sebastian Hueckesfeld, Feng Li, Casey M Schneider-Mizell, Richard D Fetter, Data curation; James W Truman, Data curation, Writing—review and editing; Albert Cardona, Conceptualization, Data curation, Writing—review and editing; Michael J Pankratz, Conceptualization, Data curation, Formal analysis, Writing—original draft, Writing—review and editing

### Author ORCIDs

Anton Miroschnikow http://orcid.org/0000-0002-2276-3434

Philipp Schlegel http://orcid.org/0000-0002-5633-1314

Andreas Schoofs http://orcid.org/0000-0001-7002-9181

Sebastian Hueckesfeld http://orcid.org/0000-0003-0236-6375

Casey M Schneider-Mizell http://orcid.org/0000-0001-9477-3853

James W Truman http://orcid.org/0000-0002-9209-5435

Albert Cardona http://orcid.org/0000-0003-4941-6536

Michael J Pankratz http://orcid.org/0000-0001-5458-6471

### Decision letter and Author response

Decision letter https://doi.org/10.7554/eLife.40247.048

Author response https://doi.org/10.7554/eLife.40247.049

## Additional files

### Supplementary files

• Supplementary file 1. PDF Neuron Atlas – Morphology and connectivity of reconstructed neurons. Reconstructions of antennal nerve (AN) sensory neurons, maxillary nerve (MxN) sensory neurons, prothoracic accessory nerve (PaN) sensory neurons, serotonergic modulatory output neurons (Se0), pharyngeal motor neurons (PMN/e .g. AN-L-motor-05), maxillary nerve motor neurons (MxN motor) and prothoracic accessory nerve motor neurons (PaN motor). A dorsal view of each neuron is shown on the left, and a lateral view on the right. Neuron IDs (e.g. '123456') and names (e.g. AN-L-Sens-B1-Aca-01) are provided. Digital 3D model of the neuropil is shown in grey. 3D models of synaptic input and output compartments are colored based on *Figure 2*. Outline of the nervous system is not shown. Table shows number of synapses of a given row neuron to a column neuron group. Column groups represent sensory neurons (ACa, AVa, AVp, ACal, ACp, ACpl, VM), neuroendocrine output neurons (Dilps, DMS, DH44), serotonergic modulatory output neurons (Se0ens, Se0ph), pharyngeal motor neurons (PMN), MxN motor neurons, PaN motor neurons and projection neurons to Kenyon cells (olfactory PNs, gustatory PNs, multiglomerular PNs, unknown PNs, thermo PNs, visual PNs).
DOI: https://doi.org/10.7554/eLife.40247.042

• Supplementary file 2. Connectivity of reconstructed neurons. Adjacency matrix with the complete connectivity of sensory neurons of the pharyngeal nerves (ACa, AVa, AVp, ACal, ACp, ACpl, VM), neuroendocrine output neurons (Dilps, DMS, DH44), serotonergic modulatory output neurons (Se0ens, Se0ph), pharyngeal motor neurons (PMN), MxN motor neurons, PaN motor neurons, alternative path interneurons (*Figure 7*), mushroom body output neurons (MBONs, MBa, MBb, *Figure 7—figure supplement 4*) and projection neurons to Kenyon cells (olfactory PNs, gustatory PNs, multiglomerular PNs, unknown PNs, thermo PNs, visual PNs).
DOI: https://doi.org/10.7554/eLife.40247.043

• Transparent reporting form
DOI: https://doi.org/10.7554/eLife.40247.044

### Data availability

All data generated or analysed during this study are included in the manuscript and supporting files. We used the same EM volume reported in Ohyama et al. 2015 (Nature) and available at https://neurodata.io/data/acardona_0111_8. To access the dataset, users need to first create a free account on the neurodata site: the data is then subsequently available to download (further details can be found in the guide https://neurodata.io/help/download/).

The following previously published dataset was used:

| Author(s) | Year | Dataset title | Dataset URL | Database and Identifier |
|---|---|---|---|---|
| Ohyama T, Schneider-Mizell CM, Fetter RD, Valdes Aleman J, Franconville R, Rivera-Alba M, Mensh BD, Branson KM, Simpson JH, Truman JW | 2015 | EM volume from: A multilevel multimodal circuit enhances action selection in Drosophila | https://neurodata.io/data/acardona_0111_8 | NeuroData, acardona_0111_8 |

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
