## [Decision Letter]

Thank you for submitting your article "Convergence of monosynaptic and polysynaptic sensory paths onto common motor outputs in a *Drosophila* feeding connectome" for consideration by *eLife*. Your article has been reviewed by three peer reviewers, Kristin Scott as Reviewing Editor and K VijayRaghavan as the Senior Editor. The following individuals involved in review of your submission have agreed to reveal their identity: Volker Hartenstein (Reviewer #2). The other two reviewers remain anonymous.

The reviewers have discussed the reviews with one another and the Reviewing Editor has drafted this decision to help you prepare a revised submission.

Summary:

The authors provide a comprehensive map of the sensory and motor elements of the *Drosophila* larval subesophageal zone (SEZ), based on serial electron microscopy. Predominantly, these elements form the input and output components of the gustatory/feeding behavior circuitry. The paper contains invaluable information for future functional studies on gustation and feeding. The authors have uncovered new and fundamental aspects of the circuitry of the feeding circuit that is dramatically different from the olfactory circuit and there are a number of surprising findings. A few concerns with nomenclature and interpretation should be addressed.

Essential revisions:

1) The entities described here, sensory organs, nerves, and sensory compartments, have been the topic of numerous previous studies, and the authors should strive to use existing names, and/or provide a look-up table where the nomenclature diverges, along with an explanation why different names were chosen.

A). The nomenclature of nerves:

There is a correspondence of larval and adult nerves. Unfortunately, in holometabolans like *Drosophila*, recognizing the continuity between adult and larval sets of nerves may require some effort, and that has led to Partially) divergent nomenclatures. That being said:

In the large majority of publications across different insect species, a pharyngeal, mandibular, maxillary and labial nerve can be distinguished for the adult, in addition to an axon bundle from the stomatogastric nervous system. In case of *Drosophila*, the mandibular nerve is reduced/cannot be distinguished; the maxillary and labial nerve are separate bundles that run close to each other/are ensheathed by a common glial envelope. Once reaching the CNS, both maxillary and labial component separate and can be clearly followed. For the peripheral nerve, the term "maxillary-labial" nerve has been used, but more commonly simply "labial nerve", or, most recently, "compound labial nerve".

In the adult *Drosophila* NS (as in all other insects), pharyngeal nerve and antennal nerve are completely separate entities; the antennal nerve is associated with the antennal segment (antennal lobe), the pharyngeal nerve with the intercalary segment (tritocerebrum). In the larva, both nerves are ensheathed by a common glial sheath (similar to maxillary/labial nerve). In some publications, the pharyngeal nerve has been called "antennal nerve" (along with the antennal nerve, which has a different origin, modality, target area, and of course has also been called "antennal nerve").

In the present manuscript, this confusion is carried forward, and further confounded.

- The labial ("compound labial") nerve is called maxillary nerve.

- The pharyngeal nerve is called antennal nerve.

- The lateropharyngeal nerve is called accessory pharyngeal nerve.

- The "pharyngeal nerve proper" plus the labial nerve (here called maxillary nerve) plus the lateropharyngeal nerve are all subsummized as "pharyngeal nerves".

The authors should either change the nomenclature, or provide an explanation/look-up table for why different terms were used.

In regard to the antennal vs ("proper") pharyngeal nerve I realize that the authors have multiple previous papers with the same word usage. Nevertheless (it is never too late to change something that is simply extremely confusing) I appeal to the logic that:

- Adult flies have clearly separate antennal and pharyngeal nerves, one to the antennal lobe (deuterocerebrum), the other to the tritocerebrum.

- In the larva, whether in the light microscope or the electron microscope, the antennal and pharyngeal component of the nerve can be clearly recognized and followed.

- The confusion extends to the motorneuron cluster that follows the pharyngeal nerve: these neurons are called "antennal (AN) motorneurons" here. These cells are all tritocerebral neruons, not antennal neurons. In other insects with large, moveable antennae, there are bonafide antennal motorneurons (innervating antennal muscles, being derived from the antennal segment. (of course, in addition to these, there are tritocerebral motorneurons, which, as in *Drosophila*, innervate the pharynx muscles. If the neurons described here remain as "antennal motorneurons", they will surely be mixed up with the "true" antennal motor neurons.

B) Nomenclature of compartments with input from defined sensilla

The authors largely use the recently introduced nomenclature for the larval and adult SEZ. Some changes and additional subdivisions are introduced, and the authors might want to explain how these changes relate to the existing map.

My understanding is that:

ACSCam corresponds to ACa (why not am?)

ACSCal ACal

ACSCp ACp

ACpl (where does this fit?)

AVSC AVa

AVSC pos AVp

VMSC VM

2) The authors define outputs to include modulatory neurons as well as motor neurons. As modulatory neurons can influence many CNS neurons and the endocrine system, it seems an unnecessarily broad grouping to consider MNs and modulatory neurons as a single class. Using different color schemes to denote modulatory versus motor, and clarifying the direct and indirect connections to modulatory versus motor would be more informative and provide a clearer picture of connectivity. Motor and modulatory should be treated as separate in the figures, text, and discussion.

3) Title and Abstract: This manuscript does not describe a feeding connectome and the title and abstract should be modified to reflect this. A feeding connectome would describe all interneurons between sensory and motor, which this study does not attempt. The title also denotes monosynaptic and polysynaptic paths to motor output which is not shown.

---

## [Author Response]

We are appreciative of the opinions of the reviewers that our work provides new and fundamental aspects of the feeding circuitry.

Essential revisions:1) The entities described here, sensory organs, nerves, and sensory compartments, have been the topic of numerous previous studies, and the authors should strive to use existing names, and/or provide a look-up table where the nomenclature diverges, along with an explanation why different names were chosen.A) The nomenclature of nerves:There is a correspondence of larval and adult nerves. Unfortunately, in holometabolans like Drosophila, recognizing the continuity between adult and larval sets of nerves may require some effort, and that has led to Partially) divergent nomenclatures. That being said:[…] In the present manuscript, this confusion is carried forward, and further confounded.- The labial ("compound labial") nerve is called maxillary nerve.- The pharyngeal nerve is called antennal nerve.- The lateropharyngeal nerve is called accessory pharyngeal nerve.- The "pharyngeal nerve proper" plus the labial nerve (here called maxillary nerve) plus the lateropharyngeal nerve are all subsummized as "pharyngeal nerves".The authors should either change the nomenclature, or provide an explanation/look-up table for why different terms were used.

We provide a new look-up table in Figure 1—figure supplement 5 which summarizes the different nomenclatures used.

In regard to the antennal vs ("proper") pharyngeal nerve I realize that the authors have multiple previous papers with the same word usage. Nevertheless (it is never too late to change something that is simply extremely confusing) I appeal to the logic that:- Adult flies have clearly separate antennal and pharyngeal nerves, one to the antennal lobe (deuterocerebrum), the other to the tritocerebrum.- In the larva, whether in the light microscope or the electron microscope, the antennal and pharyngeal component of the nerve can be clearly recognized and followed.

We are aware of the different terminologies that exist in the literature, which have come about from analysis of different developmental stages, imaging methods and comparisons to related species (as pointed out in the legend to Figure 1—figure supplement 5). In cases where we believe better clarity is achieved, we have altered our nomenclature (see below, for example). In others, we hope the new look-up reference table will act as a guide for the readers.

- The confusion extends to the motorneuron cluster that follows the pharyngeal nerve: these neurons are called "antennal (AN) motorneurons" here. These cells are all tritocerebral neruons, not antennal neurons. In other insects with large, moveable antennae, there are bonafide antennal motorneurons (innervating antennal muscles, being derived from the antennal segment. (of course, in addition to these, there are tritocerebral motorneurons, which, as in Drosophila, innervate the pharynx muscles. If the neurons described here remain as "antennal motorneurons", they will surely be mixed up with the "true" antennal motor neurons.

We have now changed AN motor neurons to PMN (pharyngeal motor neurons) all throughout the text and figures. We have also changed the abbreviation of maxillary nerve from MN to MxN, to avoid confusion with the commonly used abbreviation MN for "motor neurons" in the literature.

B) Nomenclature of compartments with input from defined sensillaThe authors largely use the recently introduced nomenclature for the larval and adult SEZ. Some changes and additional subdivisions are introduced, and the authors might want to explain how these changes relate to the existing map.My understanding is that:ACSCam corresponds to ACa (why not am?)ACSCal ACalACSCp ACpACpl (where does this fit?)AVSC AVaAVSC pos AVpVMSC VM

We have used the earlier nomenclature as a guide, but due to different methodologies used, the areas do not correspond exactly. We also did not want to use the term sensory column, since they do not look like columns, so have decided on the more neutral term sensory compartment. We have also tried to keep the abbreviations as short as possible while avoiding ambiguities, e.g., "l" for lateral was used at the end if the two hemisphere domains are clearly separated. The ACpl is a new compartment and does not correspond to earlier regions (see Figure 2—figure supplement 2B for clearer illustration of its topography).

2) The authors define outputs to include modulatory neurons as well as motor neurons. As modulatory neurons can influence many CNS neurons and the endocrine system, it seems an unnecessarily broad grouping to consider MNs and modulatory neurons as a single class. Using different color schemes to denote modulatory versus motor, and clarifying the direct and indirect connections to modulatory versus motor would be more informative and provide a clearer picture of connectivity. Motor and modulatory should be treated as separate in the figures, text, and discussion.

We are thankful for this being pointed out. We agree our phrasing was somewhat confusing in this regard. We have now gone over the text and clearly distinguished three types of output neurons: motor neurons for muscles (e.g., pharyngeal motor neurons), neuroendocrine neurons for endocrine organs (the ring gland in this case), and serotonergic modulatory neurons. We believe this has made the presentation clearer.

3) Title and Abstract: This manuscript does not describe a feeding connectome and the title and abstract should be modified to reflect this. A feeding connectome would describe all interneurons between sensory and motor, which this study does not attempt. The title also denotes monosynaptic and polysynaptic paths to motor output which is not shown.

We have now provided essentially all directly connected interneurons that lie between sensory and motor, modulatory and neuroendocrine neurons (the expanded Figure 7 and supplements). This has enabled also the connections from MBONs to interneurons and to output neurons, thus completing a circuit even from the higher centers. We think the term "a feeding connectome" can be used in this case.